# Genome biology and evolution of mating-type loci in four cereal rust fungi

**Zhenyan Luo[1], Alistair McTaggart[2], Benjamin Schwessinger[1]***

**1** Research Biology School, Australian National University, Canberra, ACT, Australia, **2** Centre for Horticultural Science, Queensland Alliance for Agriculture and Food Innovation, The University of Queensland, Ecosciences Precinct, Dutton Park, Queensland, Australia

* benjamin.schwessinger@anu.edu.au

**Data Availability Statement:** Analysis code used in this study is available at github repository: https://github.com/ZhenyanLuo/codes-used-for-mating-type. Alignments used in genealogical studies, exported RDP5 project files, ds value of gene pairs in studied cereal rust species, presumed CDS of

## Abstract

Permanent heterozygous loci, such as sex- or mating-compatibility regions, often display suppression of recombination and signals of genomic degeneration. In Basidiomycota, two distinct loci confer mating compatibility. These loci encode homeodomain (*HD*) transcription factors and pheromone receptor (*Pra*)-ligand allele pairs. To date, an analysis of genome level mating-type (MAT) loci is lacking for obligate biotrophic basidiomycetes in the *Pucciniales*, an order containing serious agricultural plant pathogens. Here, we focus on four species of *Puccinia* that infect oat and wheat, including *P. coronata* f. sp. *avenae*, *P. graminis* f. sp. *tritici*, *P. triticina* and *P. striiformis* f. sp. *tritici*. MAT loci are located on two separate chromosomes supporting previous hypotheses of a tetrapolar mating compatibility system in the *Pucciniales*. The *HD* genes are multiallelic in all four species while the PR locus appears biallelic, except for *P. graminis* f. sp. *tritici*, which potentially has multiple alleles. HD loci are largely conserved in their macrosynteny, both within and between species, without strong signals of recombination suppression. Regions proximal to the PR locus, however, displayed signs of recombination suppression and genomic degeneration in the three species with a biallelic PR locus. Our observations support a link between recombination suppression, genomic degeneration, and allele diversity of MAT loci that is consistent with recent mathematical modelling and simulations. Finally, we confirm that *MAT* genes are expressed during the asexual infection cycle, and we propose that this may support regulating nuclear maintenance and pairing during infection and spore formation. Our study provides insights into the evolution of MAT loci of key pathogenic *Puccinia* species. Understanding mating compatibility can help predict possible combinations of nuclear pairs, generated by sexual reproduction or somatic recombination, and the potential evolution of new virulent isolates of these important plant pathogens.

## Author summary

Sexes in animals and some plants are determined by sex chromosomes. In fungi, mate compatibility is determined by mating-type (MAT) loci, which share some features with sex chromosomes including recombination suppression around heterozygous loci. Here,

reconstructed HD alleles, Pra alleles, TE annotation and classification files, normalized gene expression matrix files are available at Dryad (Luo Z, Schwessinger B. Supporting information of genome biology and evolution of mating type loci in four cereal rust fungi [Dataset] 2024. Available from: https://doi.org/10.5061/dryad.w0vt4b8zm).

**Funding:** This work was supported by an Australian Research Council Future Fellowship FT180100024 to B. S. The funders had no role in study design, data collection and analysis, decision to publish, or preparation of the manuscript.

**Competing interests:** The authors have declared that no competing interests exist.

we study the MAT loci in fungal pathogens from the order *Pucciniales*, which cause rust diseases on many economically important plants, including wheat and oats. We show that one of the MAT loci is multiallelic, while the other is biallelic in most cases. The biallelic locus shows strong signs of recombination suppression and genetic deterioration with an increase in number of transposable elements and gene deserts surrounding the locus. Our findings on the genome biology of MAT loci in four economically important pathogens will improve predictions on potential novel virulent isolates that can lead to large scale pandemics in agriculture.

## Introduction

The evolutionary origin of sex and mating-type determining chromosomes or loci is a fundamental question in biology. It is widely accepted that X and Y chromosomes were originally homologous, but recombination suppression caused gradual genetic degeneration of the Y chromosome [1]. Footprints of genetic degeneration include increased rates of (non-)synonymous substitutions ($d_N$, $d_S$), accumulation of transposable elements (TEs), accumulation of inversions, reduced gene expression and/or reduced gene numbers, all of which are consequences of recombination cessation [2–5].

In contrast to many animals and plants, antagonistic selection is unlikely to be the evolutionary mechanism that causes recombination cessation in fungal mating-type (MAT) loci [6–8]. Instead, mathematical modelling and stochastic simulations suggest that non-recombining DNA fragments, caused for example by inversions, can be fixed solely due to the presence of deleterious mutations in genomes [2]. Non-recombining fragments are beneficial if they carry fewer deleterious mutations than average and can increase in frequency. However, when becoming frequent, their recessive deleterious mutations will be exposed, selected against and thereby prevent fixation. Yet, they can be fix at permanent heterozygous loci due to their sheltering effect for deleterious mutations [2].

Initial fixation of recombination suppressors, such as inversions, can lead to progressive extension of non-recombining DNA fragments by an accumulation of additional inversions over longer evolutionary timeframes [2]. Importantly, this model predicts that non-recombining DNA fragments around MAT loci are larger at biallelic loci and in species with smaller effective population sizes, short haploid phases, outcrossing mating systems, high mutation rates, and extended dikaryotic life stages [2]. Indeed, several fungal species carry extensive non-recombining DNA fragments around MAT loci. Recombination suppression can range from hundreds of kbp (kilo base pairs) to several Mbp (mega base pairs) and lead to genetic degeneration [5,9], such as in *Neurospora tetrasperma* [10], *Podospora anserina* [11], *Schizothecium tetrasporum* [12], *Agaricus bisporus* [13], *Ustilago hordei* [14], several species of *Microbotryum* [8,15] and *Cryptococcus* spp. [16,17].

Basidiomycota have evolved unique mating systems to govern nuclear compatibility, mate selection, and life cycles [9]. From a genetic perspective, two MAT loci determine mating-type identity and non-self-recognition in most Basidiomycota. The pheromone receptor (PR) locus contains a pheromone receptor gene (*Pra*) and at least one pheromone peptide precursor gene (*mfa*) [18]. The PRA protein is a transmembrane localized G protein-coupled receptor, which recognizes the processed and post-translationally modified mature pheromone peptide MFA encoded by compatible PR allele [9]. PR locus defines pre-mating compatibility and gamete fusion [3]. Downstream of initial gamete fusion, the homeodomain (HD) locus determines success in post-mating development [9]. The HD locus contains two tightly linked

homeodomain transcription factor genes (*bW-HD1* and *bE-HD2* in *Pucciniales* [19, 20], originally designated as *bE-HD1* and *bW-HD2* in *Ustilago maydis* [21]), which are linked by a short DNA fragment (~1 kbp) and are outwardly transcribed in opposite directions. Their protein products must be of different allelic specificity to form heterocomplexes and to activate a transcriptional cascade downstream of PRA-MFA [9]. The *HD* transcription factors regulate cellular development during mating, maintenance of the dikaryotic state, and control pathogenicity in some plant pathogens such as smuts [15].

The HD and PR loci can be either physically linked or unlinked, which means that they segregate together or independently, respectively. The mating compatibility system in Basidiomycota influences genomic organization and resulting segregation patterns of MAT loci. Most basidiomycete species display either bipolar or tetrapolar mating compatibility systems and sexual gametes (haploids) must have different alleles at MAT loci to be compatible [9]. In bipolar species the HD and PR loci are physically linked or alternatively, one locus has lost its function in mating compatibility. In bipolar species, only one MAT locus determines mate compatibility. In tetrapolar species, the HD and PR loci are unlinked, and both determine mate compatibility. A bipolar mating compatibility system is favorable in inbreeding populations that are primarily selfing, as the likelihood of compatibility amongst gametes derived from the same diploid individual is 50 percent [15]. In comparison, tetrapolar mating has 25 percent compatibility of gametes under selfing conditions. Outcrossing mating behavior leads to multiallelism at MAT loci because syngamy of haploid gametes derived from different individuals increases compatibility odds [6,9,15]. In many tetrapolar basidiomycete species, the HD locus is highly polymorphic within the population, with tens to hundreds of known or estimated alleles that are under negative frequency-dependent selection [15,22,23]. A tetrapolar mating compatibility system is thought to be the ancestral state in Basidiomycota, yet several species have evolved bipolarity independently. For example, *Microbotryum* spp. are highly selfing, and multiple independent evolutionary events have linked the HD and the PR locus into DNA fragments of different sizes and ages within this genus [3,7,8,24,25]. These independent events at different timescales show clear signatures of recombination suppression and genetic degeneration. Similarly, in the Ustilaginomycotina, several species evolved bipolarity by linking both MAT loci, such as *U. hordei*, *U. bromivora* or *Sporisorium scitamineum* [9]. In contrast, a recent study showed that tetrapolarity has evolved multiple times in the human skin fungus, *Malassezia* spp., where the ancestral behavior is pseudobipolar [26].

Detailed analyses of MAT loci and their genome biology is lacking for rust fungi (order *Pucciniales*, division Basidiomycota) [9]. This lack of knowledge on mating-type is despite *Pucciniales* being the largest order of fungal plant pathogens, causing diseases with significant environmental and economic impact for trees and crops such as poplar, paperbarks, wheat, oat, soybean, and coffee [27–29]. Studying mating systems in *Pucciniales* has been difficult as they are obligate biotrophs with complex life cycles. They cannot be cultured in vitro and many species require multiple hosts to complete their sexual life cycle [9]. For example, macrocyclic and heteroecious rust fungi have five spore stages and require two hosts to complete their sexual life cycles. During the asexual infection cycle on the "primary" host, for example wheat and oat for rust on cereal crops, fungi are dikaryotic and produce re-infective urediniospores that harbor two distinct nuclei carrying compatible MAT loci. The sexual cycle is initiated with the production of teliospores, which represent the diploid phase of rust fungi and the site of meiosis. Resulting haploid basidiospores infect the "alternate" host, for example *Berberis* spp. for rust fungi of cereal crops, and form haploid infection structures with distinct MAT loci. Fusion of cells with compatible MAT loci generates dikaryotic intercellular mycelia within the plant tissue that gives rise to dikaryotic aeciospores that infect the "primary" host [7] to complete the full life cycle [9, 30]. However, it is important to note that recent cytogenomic

and cytogenetic work questions this clear delineation of nuclear state and ploidy in *Pucciniales* at different life stages, with evidence for the occurrence of diploid nuclei throughout the life cycle in many species [31].

The comprehensive role of *MAT* genes during the life cycle of rust fungi is not yet understood. It is hypothesized that *MAT* genes regulate nuclear pairing during dikaryotic spore production and mediate compatibility of haploid cell fusions [9,30]. Original research suggested that most species of rust fungi are bipolar [30], yet more recent studies on *Melampsora lini* and *P. coronata* var. *coronata* revealed a tetrapolar mating compatibility system [32,33]. In the absence of direct experimental evidence such as experimental mating or gene knockout studies, genomic insights into MAT loci from genome assemblies of dikaryons can provide hypotheses on the mating compatibility system. Independent initial genome analyses of several rust fungi revealed two unlinked MAT loci with PR and HD loci located on different contigs, scaffolds or chromosomes including *Austropuccinia psidii* [34], *P. graminis* f. sp. *tritici* [35] and *P. tritici* [19,20]. This genome organization supports hypotheses of a tetrapolar mating compatibility system in these rust fungi. These rust fungi encode *Pra* receptor genes belonging to the *STE3* gene family. The likely *bona fide* receptors are encoded by *STE3.2–2* and *STE3.2–3* that display clear signatures of ancient trans-specific polymorphisms, which are proposed to be ancestral to Basidiomycota [36]. Here the sequences of a given mating-type are more similar to pheromone receptor alleles of the same mating-type in distantly related fungi than the alternate allele in the same species. In most cases, at least one *mfa* allele has been found in proximity to the STE3.2-2/3 locus. To date, the PR locus is supported as biallelic in available population level datasets for *P. tritici* and *P. striiformis* f. sp. *tritici* [19,23,37]. *Melampsora larici-populina* may be multiallelic at the PR locus [9] and population level analyses are missing for other rust fungi. The HD locus is multiallelic in *P. triticina* and *P. striiformis* f. sp. *tritici* [19,23,37] encoding variants of *bW-HD1* and *bE-HD2* that are highly dissimilar only in the variable N-terminal domain, shown to be essential for functional heterodimerization in other Basidiomycota [38,39].

It is noteworthy that direct biochemical or genetic evidence is lacking to show that any of these *MAT* genes govern mating compatibility in rust fungi. Yet, complementation assays in *U. maydis* and host induced gene silencing on wheat demonstrated the importance of *Pra* and *HD* genes of *P. triticina* for mating in a heterologous system and for spore production during asexual infection [19]. In addition, *MAT* genes were expressed during the asexual and/or sexual infection cycle in several rust fungal species [19,40]. These initial studies brought insights and left several outstanding knowledge gaps for MAT loci in rust fungi: 1) What is the chromosomal organization of PR and HD loci and how does it compare between closely related species? 2) What is the allelic diversity of *MAT* genes within rust fungal species? 3) What is the composition and organization of MAT loci proximal regions, how does it compare between PR and HD loci, and how does it vary between closely related species? 4) What is the compositional and organizational variation of MAT loci within rust fungal species? 5) Is the expression of *MAT* genes during the asexual infection cycle common in rust fungi?

Addressing these questions has been challenging because initial genome assemblies of rust fungi were fragmented and the structure of MAT loci could not be assessed consistently. For example, clustering of repetitive elements around MAT loci [18,41,42] is a challenge for studying rust fungi with repeat-rich genomes because repetitive sequences cause gaps, fragmentation, and phase-errors in assembled genomes using short-read or noisy long-read sequencing technologies. Here, we overcome this challenge and address important knowledge gaps around MAT loci in four cereal rust fungi by providing a detailed comparative analysis of publicly available chromosome-scale phased genome assemblies and complementary Illumina short-read datasets. The study species include the oat crown rust fungus, *P. coronata* f. sp. *avenae*

[43–45], and three wheat rust fungi, *P. graminis* f. sp. *tritici* [35,46,47], *P. triticina* [48–51] and *P. striiformis* f. sp. *tritici* [52–58], which combined are the biggest threat to wheat production globally causing losses of several billion dollars every year [29]. Our detailed comparative analyses set out to address the following specific objectives. 1) To determine the chromosomal organization of MAT loci. 2) To assess the allelic diversity of these loci in the four cereal rust fungi. 3) To evaluate synteny, recombination suppression, and genomic degeneration around PR and HD loci with the prediction that multiallelic loci are more syntenic than biallelic loci and that the latter show stronger footprints of recombination suppression and genomic degeneration. 4) To gauge the conservation and/or plasticity of MAT loci within a cereal rust fungus species using *P. triticina* as model system. And 5) to determine the expression of *Pra* and *HD* genes during the asexual infection cycle of the cereal hosts.

We show that *HD* genes are multiallelic in four species of cereal rust fungi and *Pra* genes are most likely biallelic in three out of four species. Only biallelic MAT loci show strong signs of recombination suppression and genomic degeneration which supports recent mathematical modelling. Our results provide novel insights into the genome biology of MAT loci in cereal rust fungi and will benefit predictions about mating and hybridization events that may be linked to the evolution of novel pathogenicity traits.

## Results

### Genomic organisation and inheritance of mating-type genes suggest cereal rust fungi have a tetrapolar mating compatibility system

We set out to test previous observations that suggest cereal rust fungi have a tetrapolar mating compatibility system with unlinked PR and HD loci [19,20]. We made use of seven available chromosome-level genome assemblies of four cereal rust fungi, namely *P. coronata* f. sp. *avenae* (*Pca*) [43–45], *P. graminis* f. sp. *tritici* (*Pgt*) [35,46,47,58], *P. triticina* (*Pt*) [37,48–51,59] and *P. striiformis* f. sp. *tritici* (*Pst*) [52,54–57,60], in addition to five partially phased assemblies (Table 1). Our initial analysis focused on one reference genome per species with *Pca 203*, *Pgt 21–0*, *Pt 76*, and *Pst 134E*.

We used previously characterized *MAT* genes from the *Pt* isolate *BBBD* (Table 2) [19] as queries to identify orthologs in available genomes and proteomes (Table 1). We identified *HD* (*bW-HD1* and *bE-HD2*) and *Pra* (*STE3.2–2* and *STE3.2–3*) alleles on chromosome 4 and chromosome 9 respectively, whereas *STE3.2–1* alleles were located on chromosome 1 (S1 Fig). This is consistent with previous preliminary analyses for *Pgt* and *Pt* that suggested HD and PR loci are located on two distinct chromosomes [35,37]. We confirmed these results using five genome resources for cereal rust fungi that are only partially phased and non-chromosome scale (Table 1). We identified two alleles of *bW-HD1*, *bE-HD2* and *Pra* (*STE3.2–2* or *STE3.2–3*) in all five genome assemblies with the exception for *STE3.2–2* in *Pst 104E* and *Pca 12SD80* [43,53]. In the case of *Pst 104E*, *STE3.2–2* was absent from the genome assembly but we confirmed its presence in the raw sequencing data by mapping sequencing reads against the *Pst 134E* phased chromosome scale genome assembly [52]. *STE3.2–2* displayed average genome-wide read coverage without any nucleotide variation confirming that *Pst 104E* was also heterozygous for *Pra*. In the case of *Pca 12SD80*, we identified two copies of *STE3.2–2* on contig 000183F and 000183F_004. This mis-characterization of *STE3.2–2* in *Pst 104E* and *Pca 12SD80* is likely caused by early long-read genome assembly errors using noisy PacBio long-reads and suboptimal genome assembly algorithms that struggle with highly repetitive regions. Similar observations have been made for missing *Pra* genes reported for earlier genome versions of *Melampsora larici-populina* [61].

**Table 1. Cereal Rust Fungi Genomes used in the present study.**

| Species | Strain | Genome size | BUSCO (%) | Source | Download date | Citation |
|---|---|---|---|---|---|---|
| *Puccinia graminis* f. sp. *tritici* | *Ug99* | 176.2 Mb | 93.4 [S:7.7, D:85.7] | NCBI | 26/05/2021 | [35] |
| | *21-0** | 176.9 Mb | 94.2 [S:8.3, D:85.9] | NCBI | 26/05/2021 | |
| *Puccinia striiformis* f. sp. *tritici* | *DK0911* | 157.8 Mb | 91.1 [S:38.4, D:52.7] | JGI | 1/06/2021 | [54] |
| | *104E* | 126.5 Mb | 92.1 [S:7.9, D:84.2] | JGI | 26/05/2021 | [53] |
| | *134E** | 167.7 Mb | 92.6 [S:6.8, D:85.8] | Provided by the authors | 21/06/2021 | [52] |
| *Puccinia coronata* f. sp. *avenae* | *12NC29* | 105.2 Mb | 89.9 [S:18.1, D:71.8] | NCBI | 26/05/2021 | [43] |
| | *12SD80* | 99.2 Mb | 90.2 [S:33.1, D:57.1] | NCBI | 26/05/2021 | [43] |
| | *203** | 208.1 Mb | 92.1 [S:6.9, D:85.2] | CSIRO data access portal | 20/06/2022 | [44] |
| *Puccinia triticina* | *76** | 253.5 Mb | 91.0 [S:1.7, D:89.3] | Provided by the authors | 28/05/2021 | [48] |
| | *15** | 243.9 Mb | 91.1 [S:1.6, D:89.5] | NCBI | 02/11/2023 | [59] |
| | *19NSW04** | 253 Mb | 91.2[S:0.9, D:90.3] | NCBI | 10/10/2023 | [37] |
| | *20QLD87** | 248.1 Mb | 91.3 [S:1.3, D:90.0] | NCBI | 10/10/2023 | [37] |
| *Puccinia polysora* f. sp. *zea* | *GD1913** | 1.7 Gb | 91.1 [S:1.4, D:89.7] | NCBI | 07/14/2023 | [81] |

The table provides information on all genome assemblies used in this study, their total dikaryotic genome assembly size ("Genome size"), and completeness as assessed with BUSCO ("BUSCO (%)"). For the "BUSCO (%)" column the first number presents the percentage of complete single BUSCO genes identified. The numbers in the brackets provide the percentage of single copy and of duplicated BUSCO genes in the dikaryotic genome assembly. Additional presented metadata includes species name ("Species"), strain name ("Strain"), source ("Source"), download date in DD/MM/YYYY ("Download date") and the initial reference ("Citation"). The * genome denotes dikaryotic genome assemblies which are chromosome scale phased genome. NCBI–National Center for Biotechnology information, JGI–Joint Genome Institute, CSIRO–Commonwealth Scientific and Industrial Research Organization.

Overall, the two MAT loci, HD and PR, are unlinked and heterozygous in all twelve dikaryotic genome assemblies, which is consistent with a genome informed hypothesis that cereal rust fungi are tetrapolar.

## *Mfa* pheromone peptide precursor genes are closely linked to *STE3.2–2* but not *STE3.2–3*

We searched the available dikaryotic genomes for putative pheromone peptide precursor genes, which encode mating-factor-a (MFA) peptides. *Mfas* are often linked to *Pra* alleles and are predicted to bind to the compatible pheromone receptors encoded at the allelic PR locus. We used the previously identified *mfa1*, *mfa2*, and *mfa3* genes of *Pst*, *Pgt*, and *Pt* as queries [19]. We identified a single *mfa2* pheromone peptide precursor gene in all species. In all cases, *mfa2* and *STE3.2–2* were closely linked, located within 500–1100 bp from each other, and encoded on the same DNA strand (S2 Fig). The *mfa2* derived precursor peptides were all 34 amino acids long with a characteristic CAAX motif at the C-terminus, where C is cysteine, A is an aliphatic amino acid, and X is any amino acid [62] (S3B Fig). The MFA2 amino acid sequences were 100 percent identical at species rank (S3B Fig).

In contrast, the sequence, length, number, and location of *mfa* genes associated with *STE3.2–3* varied between species (S2, S3A, and S3C Figs). *Pca* and *Pst* encoded a single *mfa1* allele. *Pca-mfa1* of *Pca 203* encoded a 76 amino acid peptide and was located 0.54 Mbp upstream of *STE3.2–3*. In *Pst*, *Pst-mfa1* encoded a 74 amino acid peptide and was located 10 kbp and 13 kbp away from *STE3.2–3* in *Pst 134E* and *Pst DK0911*, respectively. In contrast, *Pt* carried two identical copies of *Pt-mfa1/3* encoding 61 amino acid peptides. In the *Pt 76* reference, *Pt-mfa1/3s* were associated with *STE3.2–3* at a distance of 0.24 mbp and 0.27 mbp, respectively (S2 Fig). In addition, *Pgt* encoded for two *mfa* genes in close proximity to *STE3.2–3* that were located upstream and downstream at a distance of 30 kbp and 96 kbp, respectively

**Table 2. MAT reference genes used as query and outgroups.**

| Species | Strain | Source | Gene name | Gene ID | Citation |
|---|---|---|---|---|---|
| *Puccinia graminis* f. sp. *tritici* | isolate *CDL 75–36–700–3* (race *SCCL*) | NCBI & Cuomo et al., 2017 | *bW-HD1* | PGTG_05143 | [19] |
| | | | *bE-HD2* | PGTG_05144 | |
| | | | *STE3.2–1* | PGTG_00333 | |
| | | | *STE3.2–2* | PGTG_19559 | |
| | | | *STE3.2–3* | PGTG_01392 | |
| | | | *Pgtmfa1* | - | |
| | | | *Pgtmfa2* | - | |
| | | | *Pgtmfa3* | - | |
| *Puccinia triticina* | isolate *1–1 / race 1 (BBBD)* | NCBI & Cuomo et al., 2017 | *bW-HD1* | PTTG_09683 PTTG_27730 | [19] |
| | | | *bE-HD2* | PTTG_10928 PTTG_03697 | |
| | | | *STE3.2–1* | PTTG_09751 | |
| | | | *STE3.2–2* | PTTG_28830 | |
| | | | *STE3.2–3* | PTTG_09693 | |
| | | | *Ptmfa1/3* | - | |
| | | | *Ptmfa2* | - | |
| *Puccinia striiformis* f. sp. *tritici* | 78 | NCBI & Cuomo et al., 2017 | *bW-HD1* | PSTG_05919 PSTG_18670 | [19] |
| | | | *bE-HD2* | PSTG_05918 PSTG_19315 | |
| | | | *STE3.2–1* | PSTG_02613 | |
| | | | *STE3.2–2* | PSTG_15127 | |
| | | | *STE3.2–3* | PSTG_15070 | |
| | | | *Pstmfa1* | - | |
| | | | *Pstmfa2* | - | |
| *Puccinia polysora* f. sp. *zea* | GD1913 | NCBI | *bW-HD1* | FUNB_006397 | [81] |
| | | | *bE-HD2* | FUNA_006226 | |
| | | | *STE3.2–2* | FUNA_023003 | |
| | | | *STE3.2–1* | FUNA_001167 | |
| | | | *STE3.2–3* | FUNA_013409 | |

The table provides gene names ("Gene name") and gene identifier ("Gene ID") of *MAT* genes from *Puccinia triticina* initially used as query sequences and from *Puccinia polysora* f. sp. *zeae* used as outgroups. Additional presented metadata includes species name ("Species"), strain name ("Strain"), source ("Source"), and the initial reference ("Citation"). NCBI–National Center for Biotechnology information.

(S2 Fig). Yet in contrast to *Pt*, *Pgt mfa1* and *mfa3* encoded for highly distinct pheromone peptide precursor that were only 23.10% identical at the amino acid level and varied in length (68 vs 57 amino acids, respectively).

Further analysis of all three MFA precursor peptide alleles suggests a distinct maturation pathway for each. MFA1 precursor peptides appear to contain three tandem pheromone peptide repeats of similar sequence (S3A Fig), which has been reported for *Microbotryum* spp. [22,63] and Ascomycete fungi [64] previously. The predicted mature peptide encoded by the first repeat is sequence conserved between all four cereal rust species while the second repeat are species-specific, and the third repeat is sequenced conserved between *Pgt*, *Pst* and *Pca*, and has a single amino acid variation in *Pt* (S3A Fig). In contrast, MFA2 precursor peptides lack detectable pheromone peptide repeats and are likely processed into a single mature MFA peptide which is sequence-specific in each species (S3B Fig). Similarly, MFA3 of *Pgt* did not contain amino acid repeat sequences, however we identified the "QWGNGSHYC" amino acid sequence at the C-terminus of the Pgt-MFA3 precursor peptide (S3C Fig). This predicted mature peptide sequence of Pgt-MFA3 is highly similar to the predicted mature pheromone

peptides of Pgt-MFA1 "QWGNGSHMC" with a single amino acid variation. We cannot exclude the possibility that MFA3 is processed into additional mature peptides and future studies are needed to define if the observed amino acid variations lead to different receptor specificities.

Taken together, in all four cereal rust fungi, *Pra* alleles were associated with species-specific *mfa* genes. Yet, the number, distance, organization, sequence of *mfa* genes and their predicted mature pheromone peptides varied between species.

## Genealogies elucidate distinct evolutionary histories for the different *MAT* genes

Having identified the *MAT* genes, we investigated the genealogical relationships of each individual gene and compared them within and between species. This tested for shared or distinct evolutionary histories, which can provide indications for recombination within and between genes. For example, trans-specific polymorphisms indicate that recombination cessation is older than speciation [36,65]. In the absence of a robust phylogeny including all four study species [66], we first generated a multilocus species tree based on a multiple sequence alignment of 2,284 single ortholog protein sequences [67] including *P. polysora* f. sp. *zeae* as an outgroup (Fig 1A). The species tree showed that the three wheat rust fungi shared a most recent common ancestor when compared to the oat rust fungus *Pca* (Fig 1A). In the wheat rust clade, *Pgt* was sister to *Pt*, and *Pst* shared a common ancestor with them both. We aligned *HD* genes *bW-HD1* and *bE-HD2* separately to construct independent gene trees using *P. polysora* f. sp. *zeae* orthologs as outgroups (Table 2 and Fig 1B and 1C). The gene trees of *bW-HD1* and *bE-HD2* grouped alleles from each species into species-specific clades (Fig 1B and 1C) and we therefore did not find any evidence for trans-specific polymorphisms for either *HD* gene.

We identified multiple *bW-HD1* and *bE-HD2* alleles for the four cereal rust fungi suggesting that the HD locus is multiallelic (see also below). We observed several shared HD alleles between isolates in the case of *Pt* while HD alleles in *Pca* and *Pst* appeared to be more diverse. In the case of *Pgt*, we were limited to only four alleles, making any meaningful conclusion difficult. We next tested our null hypothesis that *bW-HD1* and *bE-HD2* have similar evolutionary histories because they are closely linked by a short DNA fragment. We performed approximately unbiased (AU) tests [68] within each species to investigate if the tree topologies of *bW-HD1* and *bE-HD2* are congruent with each other. In *Pca*, *Pst*, and *Pt* the $p_{AU}$-value was less than 0.05, which rejected the null hypothesis of congruent tree topologies. This suggests distinct evolutionary histories for the *bW-HD1* and *bE-HD2* genes in these species, which might be caused by recombination within the HD locus. In contrast, the AU-test for *Pgt* suggested similar tree topologies for *bW-HD1* and *bE-HD2*, which likely is influenced by the low sample number of four alleles. We also applied RDP5 [69] to detect potential signals of recombination at the nucleotide level. We detected potential signals for recombination in the HD locus for all species, which further supports the results of the AU test [70] and is similar to recombination events between *b* alleles reported in *U. marydis* [71].

Next, we explored the evolutionary relationship between *Pra* alleles at the PR locus by building a single gene tree using *P. polysora* f.sp. *zeae STE3.2–2* and *STE3.2–3* as outgroups. The *Pra* gene tree formed two obvious clades based on allele identity grouping the S*TE3.2–2* alleles of all species into one clade and all *STE3.2–3* alleles into another clade including respective alleles of *P. polysora* f.sp. *zeae* (Fig 2). In each clade, there was little to no intra-species variation for *STE3.2–2* and *STE3.2–3* based on the minimal branch lengths that separated allele copies in each species sub-clade. This indicates that *Pra* might be biallelic in cereal rust fungi (see also below). The clear grouping by allele identity rather than by species of rust fungi

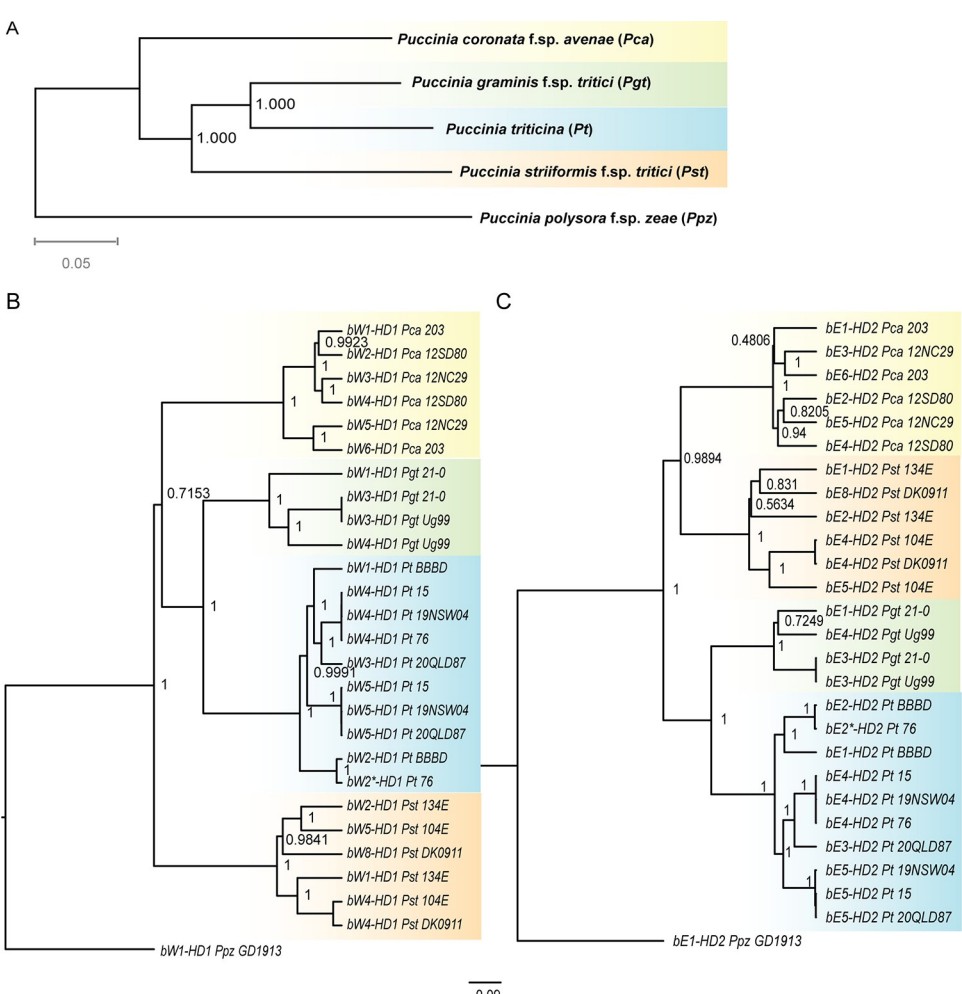

**Fig 1. *HD* genealogies suggest distinct evolutionary histories for *bW-HD1* and *bE-HD2* in four cereal rust fungi.**
(A) Species tree of *Puccinia polysora* f. sp. *zeae* (*Ppz*), *P. coronata* f. sp. *avenae* (*Pca*), *P. graminis* f. sp. *tritici* (*Pgt*), *P. triticina* (*Pt*) and *P. striiformis* f. sp. *tritici* (*Pst*) inferred from 2284 single-copy orthogroups. Numbers on nodes represent local support value computed by Fasttree. *Puccinia polysora* f. sp. *zeae* (*Ppz*) has been used for rooting the species tree. (B) and (C) Bayesian rooted gene tree built from *bW-HD1* or *bE-HD2* coding-based sequence alignment, respectively. Trees are based on a HKY + G model of molecular evolution. In either case, alleles from the same species are grouped into the same clade. Each node is labelled with its values of posterior probability (PP). PP values above 0.95 are considered to have strong evidence for monophyly of a clade and PP values of identical alleles are not displayed. The scale bar represents the number of nucleotide substitutions per site. * marks an allele with minor variations only outside the variable domain, which means this allele is predicted to be functionally equivalent to its closest neighbor. Alleles of the same species are colored with identical background: *Pca* (yellow), *Pgt* (green), *Pt* (blue), *Pst* (orange).

indicates strong trans-polymorphisms and a long-term suppression of recombination at the PR locus that predates ancestral speciation.

We next investigated if the putative pheromone peptide precursor *mfa* genes at the PR locus followed similar evolutionary histories compared to their co-located *Pra* genes. Like the *Pra* genealogy, the gene tree of *mfa* alleles revealed strong trans-specific polymorphisms because *mfa* copies clustered by allele and not species identity (S4 Fig). For example, all *mfa2* alleles grouped into one clade according to their physical association with the specific *Pra* allele *STE3.2–2*. Similarly, all *mfa1* alleles grouped together with similar topology as their linked *STE3.2–3* copies. The species-specific *Pgt-mfa3s*, which are also physically linked to *STE3.2–3*,

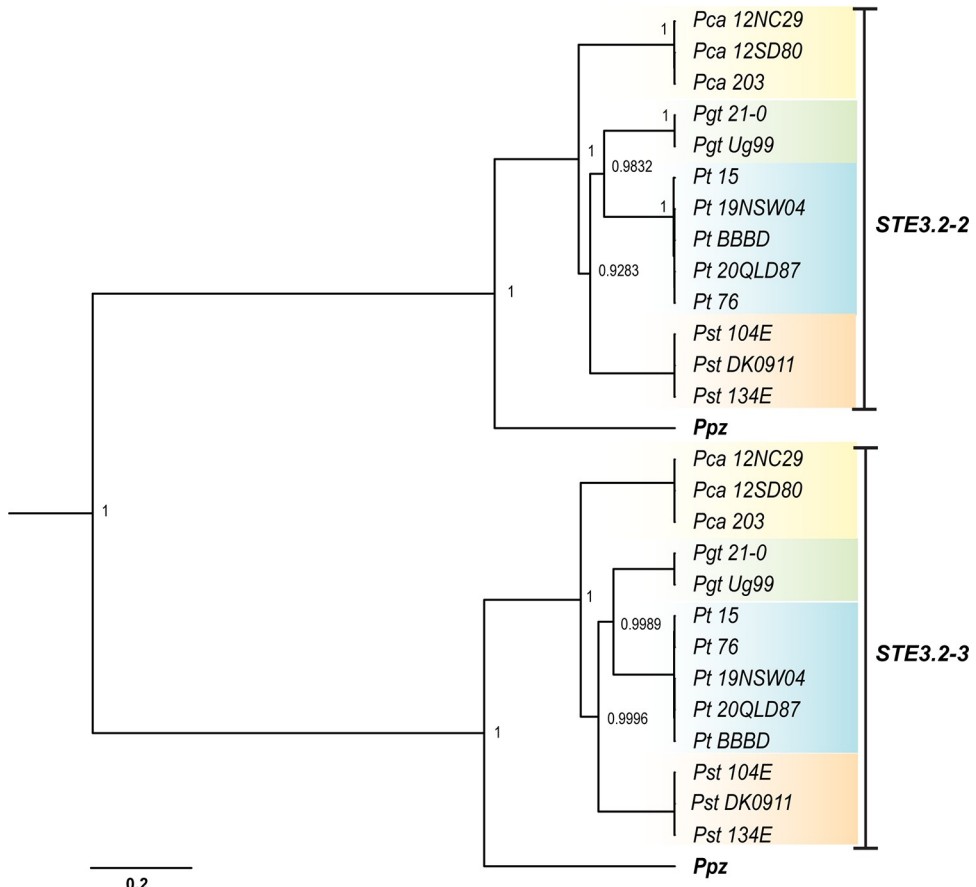

**Fig 2.** *Pra* **genealogy displays ancient trans-specific polymorphism in cereal rust fungi.** Bayesian gene tree built from *Pra* (*STE3.3–2* and *STE3.3–3*) coding-based sequence alignment. The tree is based on a TN93 + I model of molecular evolution. The *Pra* alleles, *STE3.2–2* and *STE3.2–3*, were grouped into two clades by allele identity and not species identity. Each node is labelled with its values of posterior probability (PP). PP values above 0.95 are considered as strong evidence for monophyly of a clade and PP values of identical alleles are not displayed. The scale bar represents the number of nucleotide substitutions per site. Species-specific background coloring is the same as for Fig 1.

were placed as a sister group to the *mfa1* clade, being more closely related to *mfa1* than to *mfa2*.

## HD loci display limited signs of genomic deterioration and their synteny is conserved within species

Different effects on recombination have been reported around MAT loci in basidiomycetes [8,17]. Hence, we investigated signals of altered recombination, synteny, and genomic deterioration around the MAT loci in rust fungi. We defined the HD locus as the DNA fragment which includes *bW-HD1*, *bE-HD2*, and the DNA sequence in between [22]. First, we analyzed whether there was a reduction in synteny on chromosome 4, surrounding the HD locus, as a signature for long-term recombination suppression. We used MUMmer software [72] to align the haplotypes of chromosome 4 from *Pca*, *Pgt*, *Pt*, and *Pst*, against each other (S5 Fig). Overall macro-synteny of chromosome 4 haplotypes was conserved in all cereal rust fungi with *Pt 76* the most syntenic (S5C Fig). Macro-synteny in a 40 kb-sized window around the HD locus was mostly conserved in *Pt 76*, *Pca 203*, and *Pst 134E* while this initial analysis suggested that this locus was less syntenic in *Pgt 21–0* (S5 Fig).

We investigated the individual allelic divergence along chromosome 4 using synonymous divergence ($d_S$) value as an additional measure for footprints of ancient recombination suppression [8]. This analysis tested if the HD locus and proximal genes showed an increase in $d_S$ values, which could be caused by long-term suppression of recombination and accumulation of independent mutations, as seen in other Basidiomycota [15]. We calculated the pairwise $d_S$ values for all genes across all sister chromosomes in all four species of *Puccinia*. This whole-genome analysis of all allele pairs on sister chromosomes suggested that chromosome 4 did not have increased $d_S$ values when compared to all other chromosomes (S6 Fig). The $d_S$ value of alleles for both *HD* genes fell within the upper 95% quantile of $d_S$ values on chromosome 4. Yet, $d_S$ values of directly adjacent genes were not elevated beyond the background level when we plotted $d_S$ values for allele pairs along chromosome 4 (Fig 3). Consistently, regions around the HD locus did not show any obviously particular patterns of gene or transposon density, which are common features of genetic deterioration (Fig 3). We tested whether the HD locus was potentially linked to centromeres as reported in other Basidiomycota [15,73]. We used previously identified centromere locations for *Pgt* and *Pst* [52,74] and identified centromere locations for *Pca* and *Pt* using genome-wide Hi-C heatmaps based on the bowtie-like Hi-C interaction features known for fungal centromeres (S7–10 Figs) [75]. This analysis revealed that the HD locus is likely not directly linked to centromeres in any of the four cereal rust fungi.

Lastly, we investigated overall conservation of nucleotide and gene-coding regions for the HD locus including proximal regions containing 40 neighboring genes on either side of the locus. We combined fine scale nucleotide synteny with gene conservation analysis. We specifically tested if cereal rust fungi contain syntenic blocks with conserved protein-coding genes surrounding MAT loci as was found for species e. g. *Trichosporonales* spp. [17]. We used blastn to identify conserved nucleotide sequences within dikaryotic genomes and between species. We plotted genes, transposons, nucleotide and protein-coding gene conservation for proximal regions to the HD locus for all four cereal rust fungi (Fig 4A). Protein-coding genes and their order are mostly conserved within dikaryotic genomes of the same species with 58/80 genes in *Pca 203*, 40/80 in *Pgt 21–0*, 57/80 in *Pt 76*, and 62/80 in *Pst 134E* being conserved. Similarly, we observed considerable nucleotide conservation and synteny within dikaryotic genomes with the exception of *Pgt 21–0*, which is consistent with our initial MUMmer-based analysis (S5 Fig). Overall, synteny and protein-coding gene conservation was very limited between species for regions proximal to the HD locus. We could only identify three conserved genes across all four cereal rust fungi. These three genes code for an integral membrane protein (PTHR12459), D-2-hydroxyglutarate dehydrogenase (PTHR43716) and Glucose-6-phosphate 1-sepimerase (IPR025532) with all three clustering downstream of the HD locus (Fig 4A). Our TE analysis revealed that their coverage was overall consistent at the order classification level [76] within species but varied across species (S11 Fig). For example, terminal inverted repeats (TIRs) and other undetermined Class II DNA transposons dominated the HD locus of *Pst 134E*, while the HD locus of other rust fungi had higher coverage of Class I RNA transposons including long terminal repeats (LTRs).

Overall, our analysis of the HD locus suggests that it is conserved within each dikaryotic genome at species rank, while conservation has been eroded between species over longer evolutionary timescales.

## The PR locus displays strong signs of genomic degeneration

We investigated organization and synteny of the PR locus which we defined as the DNA fragments including *Pra*, *mfa* genes, and DNA sequences in between. Initial MUMmer based

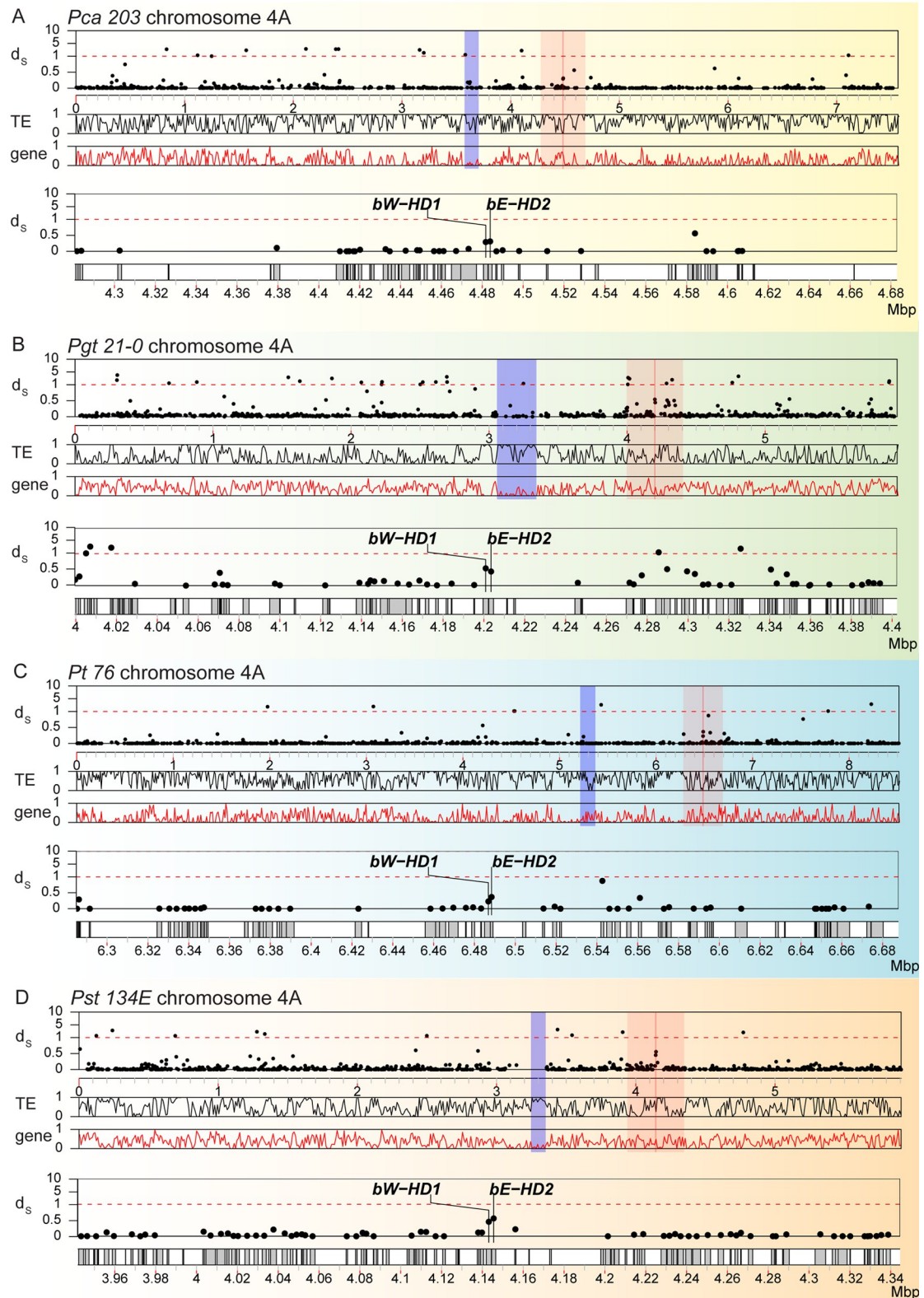

**Fig 3. Synonymous divergence (d$_S$) values of *HD* genes but not their immediate neighbors are slightly elevated on chromosome 4.** Synonymous divergence values (d$_S$) for all allele pairs are plotted along chromosome 4A for (A) *P. coronata* f. sp. *avenae* ("*Pca 203*"), (B) *P. graminis* f. sp. *tritici* ("*Pgt 21–0*"), (C) *P. triticina* ("*Pt 76*"), and (D) *P. striiformis* f. sp. *Tritici* ("*Pst 134E*").

In each panel, the top track shows the $d_S$ values ("$d_S$") of allele pairs along chromosome 4. Each dot corresponds to the $d_S$ value of a single allele pair. The second and third track shows the averaged TE ("TE") and gene ("gene") density along chromosome 4 in 10 kbp-sized windows, respectively. The *HD* genes (*bW-HD1* and *bE-HD2*) are highlighted with a red line and red shading indicates a 0.4 mbp-sized window around the HD locus. Predicted centromeric regions are marked with blue shading. The two lower tracks ($d_S$ values and gene locations) provide a detailed zoomed in view of red shaded area around the HD locus. Species-specific background coloring is the same as for Fig 1.

synteny analysis revealed clear macro- and micro-synteny breaks around the PR locus in all cereal rust fungi (S12 Fig). It also revealed accumulation of highly repetitive sequences closely associated with the PR locus, especially in *Pca 203* and *Pst 134E*. This increase in TEs correlates with a reduction in gene density around the PR locus (Figs 4B and 5). The loss in synteny was specific to the PR locus, as overall macro-synteny of chromosome 9 was conserved in all species (S12 Fig).

Consistent with the macro-synteny analysis, $d_S$ values for the *Pra* alleles *STE3.2-2- STE3.2–3* belonged to the 99.9% quantile of $d_S$ values of all allele pairs found on chromosome 9 in all species (Figs 5 and S6). Gene density dropped significantly around *Pra* genes with *STE3.2-2/ STE3.2–3* located within or adjacent to 'gene deserts' that extended over 1 mbp in the case of *Pst 134E*. The PR locus and its proximal regions also appear to be enriched in TEs. One exception was observed around the PR locus of *Pgt 21–0* which exhibited a higher gene density and lower amounts of TEs when compared to the other three species, which might be a biological feature or a technical artefact (see Discussion). None of the *Pra* alleles was physically closely linked to centromeres except for *Pst 134E STE3.2–3*, which was found 164 kbp from the centromere, embedded in a > 1 Mb long stretch of TEs depleted of genes (Fig 5).

Detailed analyses of nucleotide synteny and protein-coding gene conservation confirmed signals of extended recombination suppression, loss of synteny and accumulation of transposable elements around the PR locus (Fig 4B). We observed little conservation of nucleotide sequences and protein-coding genes in regions proximal to *Pra* within dikaryotic genomes or between species. Indeed, we could not identify any conserved protein-coding gene proximal to *Pra* across the four cereal rust fungi. The TE composition in regions proximal to *Pra* varied within dikaryotic genomes and between species (S11B Fig). This indicates the accumulation of distinct transposable element families around the PR locus in different cereal rust species (see more detail for *Pt* below).

Our results show that the PR locus appears highly plastic with little conservation within and between species.

## *HD* genes are multiallelic in cereal rust fungi while *Pra genes* display little variation at the species rank

We extended our species rank analysis of variation in *MAT* genes (Figs 1 and 2) using publicly available genomic data for all four species (S1 Table). We identified between 12 or 15 representative isolates per species that were assigned to as many distinct genomic lineages as possible based on previous population genetic studies [35,45–47,49–51,55–58] (S1 Table). Our reasoning for this approach was to avoid repeated sampling from clonal lineages that are predicted to show no genetic variation at *MAT* genes. We quality controlled whole-genome short-read datasets and mapped them against the respective reference genome to estimate the genetic variation in *MAT* genes (S13–S16 and S18–S21 Figs). We identified high levels of polymorphisms at the HD locus for all analyzed cereal rust fungi. In our dataset, *Pca* showed the highest level of variation of the HD locus for all species

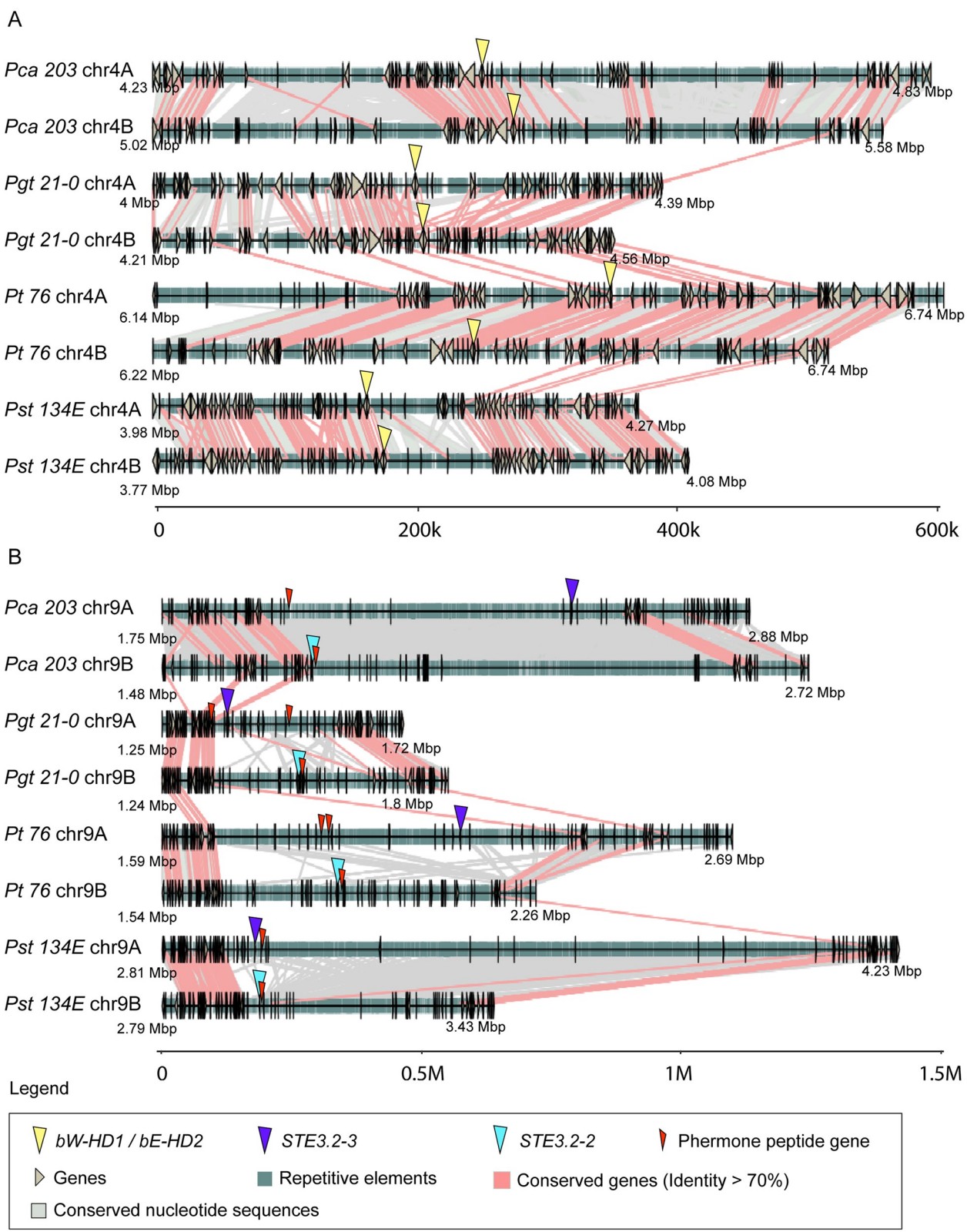

**Fig 4. HD loci are partially conserved within species whereas PR loci are highly heterozygous and display strong signals of genomic degeneration.** Synteny graphs of HD locus (A) and (B) PR locus including proximal regions in *P. coronata* f. sp. *avenae* ("*Pca 203*"), *P. graminis* f.

sp. *tritici* ("*Pgt 21–0*"), (C) *P. triticina* ("*Pt 76*"), and (D) *P. striiformis* f. sp. *tritici* ("*Pst 134E*"). Proximal regions are defined as 40 genes downstream and upstream of *HD* and *Pra* genes, respectively. HD loci are relatively syntenic within each dikaryotic genome but less conserved between species. PR loci display little synteny in *Pgt 21–0*, *Pt 76* and *Pst 134E* and show strong signs of transposon accumulation (see also S13 Fig). There is very little conservation of the PR loci across species. Red lines between chromosome sections represent gene pairs with identity higher than 70% and grey shades represent conserved nucleotide sequences (> = 1000 bp and identity > = 90%). For additional annotations please refer to the included legend ("Legend").

(S13 Fig). When mapping Illumina short-read data of the *Pca* isolates against the *Pca 203* reference, the HD locus was highly polymorphic including heterozygous single nucleotide polymorphisms (SNPs) and multiple unaligned regions and gaps. This suggested that the HD locus present in our selected *Pca* isolates were not well reflected in the *Pca 203* reference assembly. This is likely due to sampling from diverse sexual populations of *Pca* in the USA [45]. We therefore implemented a novel *de novo* reconstruction approach for the HD locus its encoded *HD* genes directly from whole-genome Illumina short-read datasets (see Methods for detail). We confirmed our approach by reconstructing the *Pca 203* HD locus from publicly available Illumina short-read datasets. We confirmed that the two reconstructed *HD* alleles were identical to the ones derived from the dikaryotic reference genome assembly (Fig 1) via a nucleotide alignment of the two reference *bW-HD1* and *bE-HD2* alleles to the *de novo* reconstructed *HD* alleles (S17A Fig). In total, we reconstructed eleven *bW-HD1* and twelve *bE-HD2* alleles for *Pca* (S17B Fig). Using this approach, we identified extensive genetic variation for *bW-HD1* and *bE-HD2* in *Pgt* and *Pt* (S17C–S17D Figs) while making use of recently published *bW-HD1* and *bE-HD2* alleles in case of *Pst* [23] (S17E Fig). We identified six, nine, and ten *bW-HD1* and six, eight, and ten *bE-HD2* alleles for *Pgt*, *Pt*, and *Pst*, respectively. Overall, this suggests that the *HD* genes are multiallelic in all four cereal rust fungi, which is consistent with other recent reports [23,37].

Compared to the *HD* genes, the *Pra* genes are reported to be far less polymorphic in many basidiomycetes [9,77,78]. Hence, we next focused our analysis on *Pra* genes using the same whole-genome Illumina short-read datasets (S1 Table). As expected, the *Pra* genes were far less polymorphic within each species compared to *HD* genes (S18–S22 Figs), which is consistent with our initial analysis of *Pra* genes extracted from genome assemblies (Figs 1 and 2). In contrast to the highly polymorphic *HD* genes in *Pca*, we identified only two SNPs in the coding regions of *STE3.2–2* and *STE3.2–3* with a single SNP being non-synonymous (S22 Fig). Similarly, *STE3.2–2* and *STE3.2–3* copies of all *Pst* isolates had identical coding sequences. Our short-read mapping analysis of *Pra* in *Pt* identified four SNPs in coding regions of *STE3.2–2* and *STE3.2–3*. These SNPs gave rise to a single additional *STE3.2–2* variant with two amino acid substitutions close to the C-terminus (S22 Fig). *Pgt* was the most polymorphic for *Pra*. We identified multiple distinct copies of *STE3.2–2* and *STE3.2–3* in the global *Pgt* population. We identified one *STE3.2–2* variant with several amino acid changes (S22B Fig). *STE3.2–3* was the most polymorphic in *Pgt* including two isolates, *TTTSK* and *UVPgt60*, which contain potential non-sense mutations leading to pre-mature stop codons (S22B Fig). Overall, our analysis suggests that *Pra* is most likely biallelic in *Pca*, *Pt*, and *Pst*, while *Pgt* may have more than two functional *Pra* alleles.

Lastly, we investigated the nucleotide sequences of all *mfa* alleles at the species rank. *mfa1* and *mfa2* alleles were fully conserved in *Pca* and *Pst*, while *Pt* had only a single non-synonymous change in *mfa1/3* (S3 Fig). In contrast, *mfa1* and *mfa3* in *Pgt* had several non-synonymous variations, yet all were located outside the predicted mature pheromone peptide sequences (S23 Fig). This species level variation in *Pgt* at *mfa* is consistent with the variation observed in *STE3.2–2* and *STE3.2–3*.

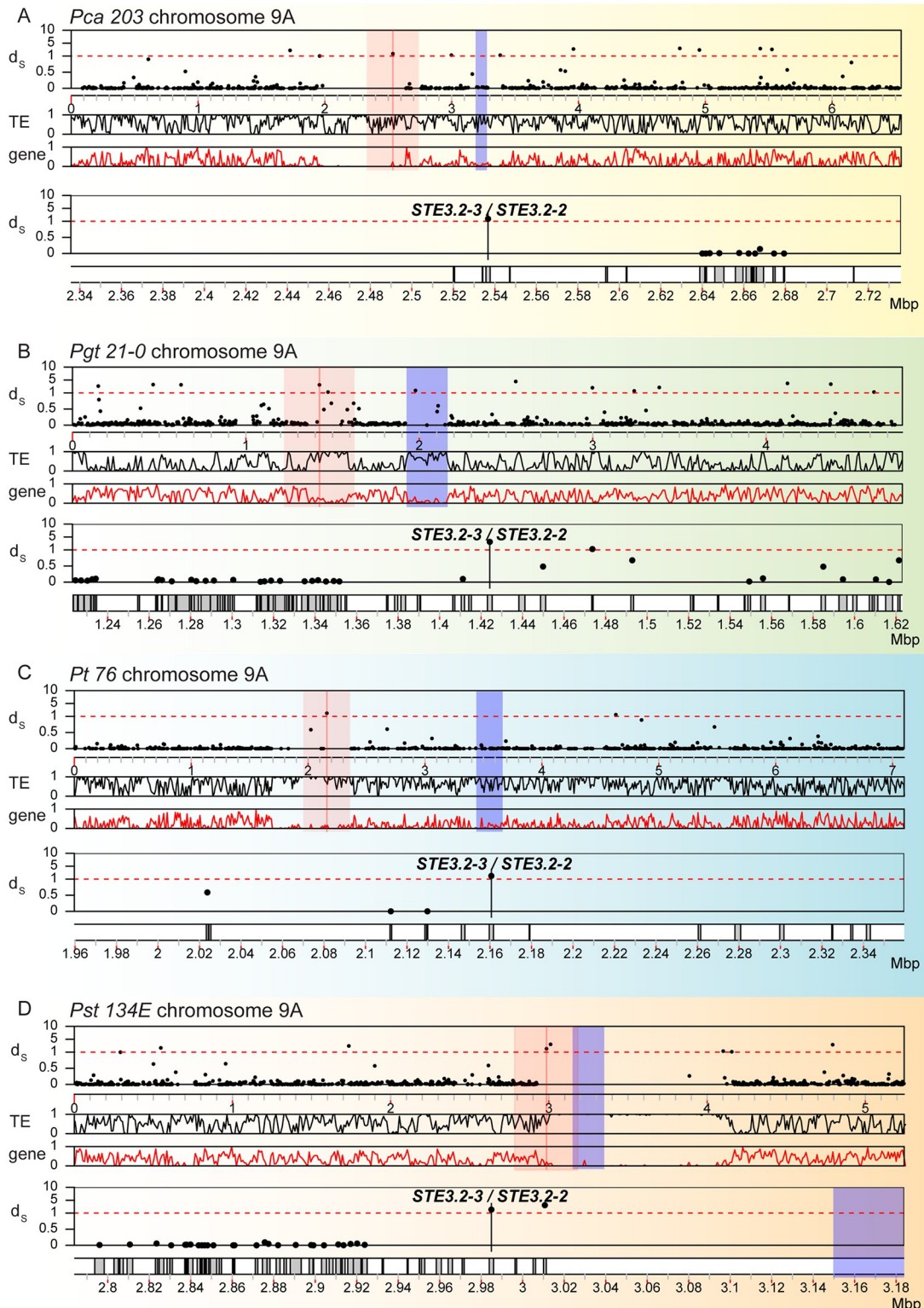

**Fig 5. PR loci display elevated synonymous divergence (d_S) values, accumulation of transposable elements and depletion of genes.** Synonymous divergence values (d_S) for all allele pairs are plotted along chromosome 9A for (A) *P. coronata* f. sp. *avenae* ("*Pca 203*"), (B) *P. graminis* f. sp. *tritici* ("*Pgt 21–0*"), (C) *P. triticina* ("*Pt 76*"), and (D) *P. striiformis* f. sp. *tritici* ("*Pst 134E*"). In each panel, the top track shows the d_S values ("d_S") of allele pairs along chromosome 9. Each dot corresponds to the d_S value of a single allele pair. The second and third track shows the averaged TE ("TE") and gene ("gene") density along chromosome 9 in 10

kbp-sized windows, respectively. The *Pra* alleles (*STE3.2–2* and *STE3.2–3*) are highlighted with a red line and red shading indicates a 0.4 mbp-sized window around the PR locus. Predicted centromeric regions are marked with blue shading. The two lower tracks (d$_S$ values and gene locations) provide a detailed zoomed in view of red shaded area around the PR locus. Species-specific background coloring is the same as for Fig 1.

### MAT loci are conserved at the species level in *Puccinia triticina* while showing transposable element expansion specific to the *STE3.2–3* allele

We made use of four chromosome-scale and fully-phased genome assemblies for *Pt* (Table 1) to explore structural conservation or plasticity of the MAT loci within a species of cereal rust fungi. We analyzed macro-synteny, detailed nucleotide synteny, protein-coding gene conservation and TE composition for the HD and PR locus within and between *Pt 76* [48], *Pt 15* [59], *Pt 19NSW04* and *Pt 20QLD87* [37]. A recent study suggested that *Pt 19NSW04* arose via somatic hybridization between *Pt 76* and *Pt 20QLD87* or close relatives in Australia mid-2010 [37]. Hence, *Pt 19NSW04* and *Pt 76* share one complete nuclear haplotype B, which contains *STE3.2–2*, and *Pt 19NSW04* and *Pt 20QLD87* share one complete nuclear haplotype C, which contains *STE3.2–3*. The *Pt 15* isolate is unrelated and was sampled in China in 2015.

Macro-synteny plots of the HD locus containing chromosomes suggested that shared nuclear haplotypes e.g. *Pt 19NSW04* and *Pt 76* hapB or *Pt 19NSW04* and *Pt 20QLD87* hapC, are more conserved when compared to the HD locus contained within each dikaryotic genome (S5 and S24 Figs). Detailed nucleotide synteny and protein-coding gene conservation analysis revealed that gene order and function is highly conserved in regions proximal to the HD locus between all tested nuclear haplotypes (S25 Fig). To investigate whether any specific TE families cluster around the HD locus, we built a TE database based on the *Pt 76* reference genome and used it to annotate TEs in all four isolates. We identified similar coverage of TEs, at the order classification level, around the HD locus in all dikaryotic genomes (S26 Fig).

We used identical analyses for the PR locus to understand its evolution at species level in *Pt*. We compared the overall structural relationship between chromosomes containing *STE3.2–2* or *STE3.2–3* using macro-synteny plots. Like the HD locus, chromosomes of shared nuclear haplotypes, e.g. *Pt 19NSW04* vs *Pt 76* hapB with *STE3.2–2*, were more similar to each other than non-shared haplotypes, e.g. *Pt 76* hapB vs *Pt 15* hapA with *STE3.2–3* (S12 and S27A Figs). The same applied for *STE3.2–2* (S12 and S27B Figs). We clearly observed clustering of repetitive sequences around the *STE3.2–2* and *STE3.2–3* locus with the latter more pronounced (S27 Fig). These repetitive sequences appeared to be allele specific as they were not visible in macro-synteny plots comparing *STE3.2–2* and *STE3.2–3* containing chromosomes (S12 Fig). Detailed analyses of nucleotide synteny and protein-coding genes showed that genes proximal to either *Pra* allele were mostly conserved while some haplotypes appeared to have inversions downstream of *STE3.2–2* relative to the *Pt 76* reference (Fig 6A and 6B). Gene synteny was more conserved for genes surrounding *STE3.2–3* yet gene distance appeared to be especially variable between the two *mfa1/3s* and *STE3.2–3* (Fig 6B). We explored the TE composition around each *Pra* allele and tested if TE expansions could explain the difference in intergenic distances around the *STE3.2–3* allele. TE composition around *STE3.2–2* and *STE3.2–3* were markedly different with the *STE3.2–3* allele displaying an increase of LTR retrotransposons (Fig 6C). TE coverage and composition around *STE3.2–2* was overall consistent between the different haplotypes (Fig 6C). In the case of *STE3.2–3* we found a single LTR *Ty3* (also known as Gypsy [79]) TE family (Ty3_Pt_*STE3.2–3*, Fig 6D) highly expanded at the locus with varying coverage between haplotypes ranging from 14.7% to 25.4% with *Pt 20QLD87* having the highest coverage (Fig 6C). The percentage identity relative to the consensus sequence was > 99% and most copies were around 8 kb (Fig 6E), which indicates active transposition of

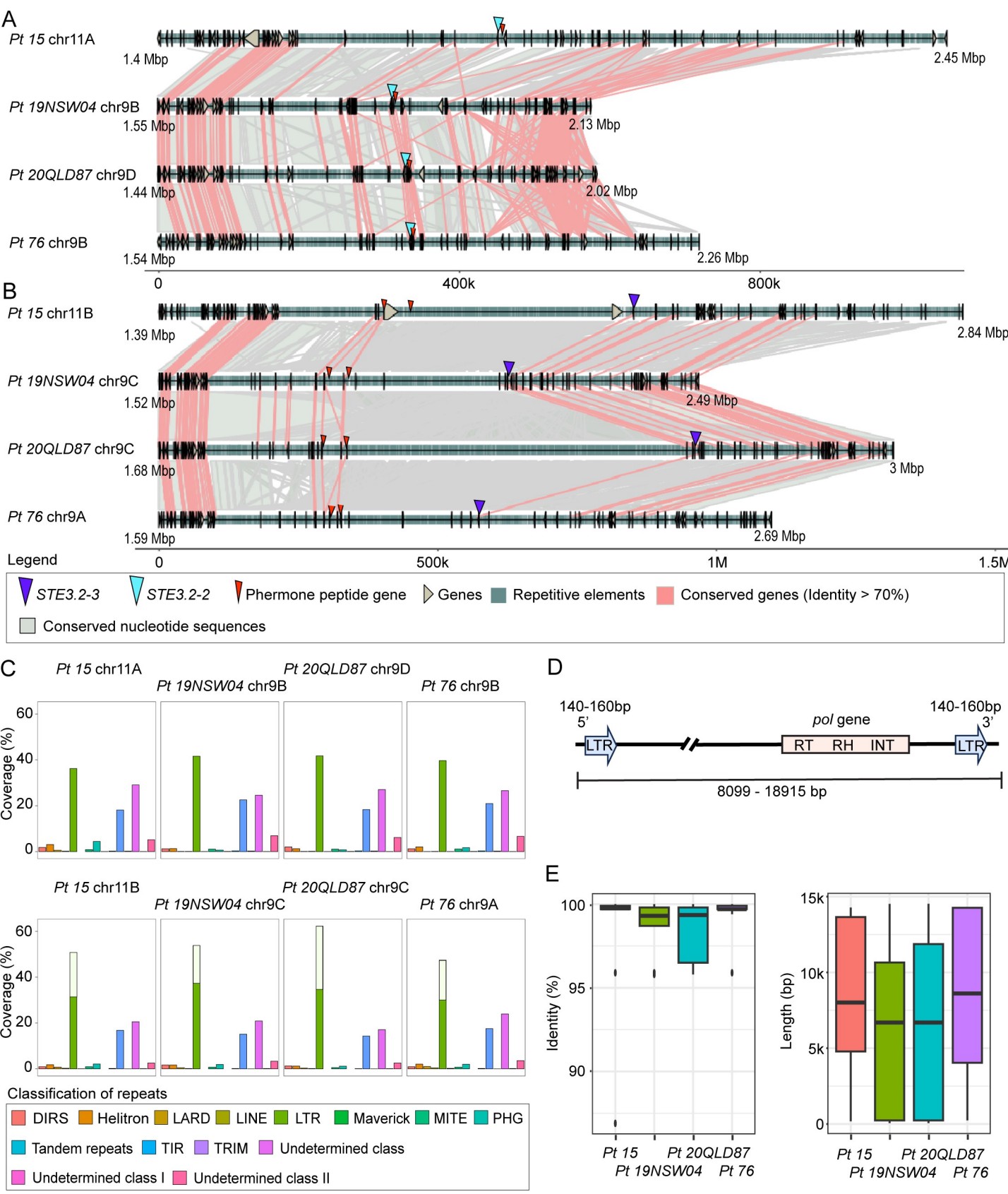

**Fig 6. The *PR* locus shows an *STE3.2–3* allele specific accumulation of a unique Ty3-transposon family in *Puccinia triticina*.** Synteny graphs of (A) *STE3.2–2* and (B) *STE3.2–3* including proximal regions in four different *P. triticina* isolates *Pt 15*, *Pt 19NSW04*, *Pt 20QLD87*, and *Pt 76*. Both loci are overall syntenic while the *STE3.2–3* locus displays an extension of the intergenic distance especially between the pheromone receptor and the two *mfa* genes. Red lines between chromosome sections represent gene pairs with identity higher than 70% and grey shades represent conserved nucleotide sequences (> = 1000 bp and identity > = 90%). For additional annotations please refer to the included legend ("Legend"). (C) Transposable element coverage at the order classification level at the PR locus in the four different *P. triticina* isolates, empty white bars represent the coverage of Ty3_Pt_*STE3.2–3*. (D) Structure of a representative Ty3_Pt_*STE3.2–3* copy. LTR–Long Terminal Repeat, RT–Reverse transcriptase, RH–RNAse H, and INT–Integrase. Distributions of (E) percentage identity relative to the consensus sequence (left) and lengths (right) of individual Ty3_Pt_*STE3.2–3* insertions at the *STE3.2–3* gene and proximal regions in four different *P. triticina* isolates.

Ty3_Pt_*STE3.2–3* in recent history. To investigate whether this TE family was specific to the *STE3.2–3* allele, we compared its abundance at the PR locus with its abundance across the whole-genome. We found that Ty3_Pt_*STE3.2–3* only clustered on *STE3.2–3* containing chromosomes in all four strains with only a very limited number of single copies found on other chromosomes (S28 Fig). Indeed, Ty3_Pt_*STE3.2–3* had a strong preference to insert between *Ptmfa1/3s* and *STE3.2–3* (S29 Fig). To further investigate the structure of Ty3_Pt_*STE3.2–3*, we used LTR_FINDER [80] to identify 5' and 3' LTR regions and other features related to retrotransposons. Ty3_Pt_*STE3.2–3* was found to have a short LTR, around 150 bp in all four strains, with target site repeat (TSR) region of 'AAGT', *pol* genes containing reverse transcriptase (RT), ribonuclease H (RH), integrase (INT). However, we failed to identify any group-specific antigen (*gag*) in these TEs (Fig 6D).

## *STE3.2–1* genes are highly conserved within and among cereal rust fungi

Similar to previous studies [19,34], we identified an additional *Pra*-like gene, *STE3.2–1*, on chromosome 1. However, its two alleles are nearly identical in all dikaryotic genome assemblies and are not associated with any pheromone peptide precursor genes [70]. We investigated the genealogy of the *Pra*-like *STE3.2–1* with using *P. polysora* f. sp. *zeae STE3.2–1* as an outgroup [81]. The two *STE3.2–1* alleles identified in each isolate were almost identical and grouped by species identity. Overall, *STE3.2–1* showed very little variation within each species (S30 Fig).

We also analyzed *STE3.2–1* and its proximal regions as above for the MAT loci. The three independent analyses revealed that *STE3.2–1* and its proximal regions are highly conserved at the nucleotide and protein-coding gene level within dikaryotic genomes and are syntenic between species (S31–S33 Figs). This shows a clear difference between the *STE3.2–1* gene from the two *MAT* genes. Hence, *STE3.2–1* is likely not involved in mating compatibility in the examined cereal rust fungi as previously suggested.

## *MAT* genes are expressed late in the asexual infection cycle during urediniospore production

Importance of *MAT* genes in mate compatibility is well established for *Pt* and *Pst* [19,82], yet it is unclear if they are expressed and functional during asexual reproduction on cereal hosts as found in *Pt* [19]. Hence, we investigated the expression patterns of *MAT* genes in *Pca*, *Pgt* and *Pst* using publicly available RNA-seq infection time series of their cereal hosts [43,46,53,83] (S2 Table). We applied the trimmed mean of M-values (TMM) normalization to read counts and assessed quality of the RNA-seq datasets by multidimensional scaling (MDS) plots (S34 Fig). The MDS plots confirmed the suitability of the datasets for detailed expression analysis based on technical replicates clustering closely together and one dimension separating samples according to their infection progress. The stable expression of two house-keeping genes for each species across all samples further confirmed the suitability of the datasets for detailed *MAT* gene expression analysis (S35 Fig).

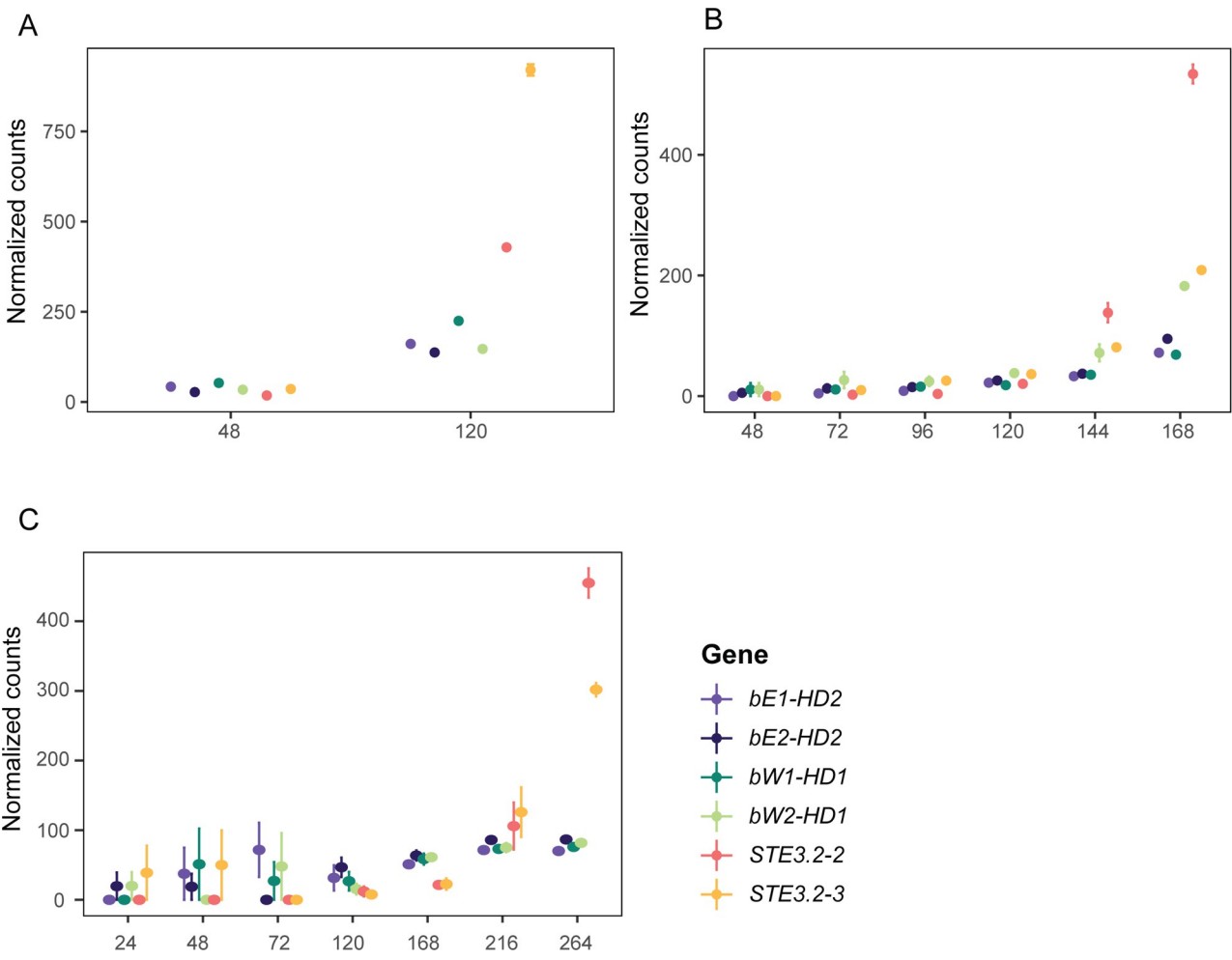

**Fig 7. *MAT* gene expression is upregulated in the late stage of asexual infection cycle of the cereal host.** (A) Trimmed mean of M-values (TMM)-normalized values of *MAT* genes in *P. coronata* f. sp. *avenae* ("*Pca 12NC29*") at 48 and 120 hour post infection (hpi). (B) TMM-normalized values of *MAT* genes in *P. graminis* f. sp. *tritici* ("*Pgt 21–0*") at 48, 72, 96, 120, 144 and 168 hpi. (C) TMM-normalized values of *MAT* genes in *P. striiformis* f. sp. *tritici* ("*Pst 87/66*") at 24, 48, 72, 120, 168, 216 and 264 hpi.

*STE3.2–2* and *STE3.2–3* were upregulated during the asexual infection process and always displayed highest expression at the latest time point available, which coincides with sporulation and production of urediniospores (Fig 7). We confirmed the differential expression at later infection timepoints with a likelihood ratio test with a p-value cut-off of < 0.05 (S36 Fig). Similarly, *HD* genes are upregulated during asexual infection of the cereal host (Fig 7). We did not observe any expression of *STE3.2–1* in any of the samples. Lastly, we were also interested to see if we could detect transcripts of *MAT* genes in cereal rust spores before infection and in specialized infection structures called haustoria. We used an additional *Pst* RNA-seq dataset and compared the expression of *MAT* genes in ungerminated spores, germinated spores, 5-days post infection, 9-days post infection and haustoria enriched samples (S37 Fig) [53]. *HD* genes displayed some expression in spores albeit lower than observed in later stages of infection during spore production in the asexual cycle. In contrast, we did not detect any expression of *Pra* genes in ungerminated and germinated spores. Yet, *Pra* genes were expressed and upregulated at later stages of the asexual infection consistent with other publicly available datasets (Figs 7C and S37).

## Discussion

Our comparative analysis of predicted *MAT* genes and MAT loci in four cereal rust fungi provides novel insight into the evolution of these genes and their proximal regions within and between species.

First, we confirmed that HD and PR loci are unlinked and located on two distinct chromosomes in all four cereal rust fungi, which supports previous analyses on more fragmented genome assemblies or for individual rust fungal species [19,23,35]. Further, both loci were heterozygous in all twelve dikaryotic genome assemblies. Taken together, these observations strongly support the hypothesis of rust fungi displaying a tetrapolar mating compatibility system. Though this genome biology informed hypothesis requires experimental validation via crosses and genetic analysis of resulting offspring.

As previously observed for *Pt* [19], the *Pra* homologs *STE3.2–1* were highly conserved in all cereal rust fungi and had no associated *mfa* genes. This suggests that *STE3.2–1* does not regulate mate compatibility. In *P. coronata* f. sp. *avenae*, *P. striiformis* f. sp. *graminis* and *P. striiformis* f. sp. *tritici*, *STE3.2–1* is flanked by anaphase promoting complex *APC10/Doc1*, which regulates mitosis [84], and Elongation factor *EF1B* [85]. Together, these findings suggest that *STE3.2–1* might play a role during the cell cycle or different developmental programs as reported for other basidiomycete fungi [86,87]. However, as previously reported for *P. triticina* [19], expression for *STE3.2–1* was absent or very low during the infection cycle of the cereal host.

Fungi with a tetrapolar mating compatibility system are often multiallelic for at least one of the two MAT loci. For example, *U. maydis* has many distinct transcription factor alleles at the HD locus [88,89]. We performed extensive analysis of the HD locus extracted from dikaryotic genome assemblies and developed a workflow to *de novo* reconstruct loci directly from Illumina short-read sequencing data. We investigated the allelic diversity of *bW-HD1* and *bE-HD2* in 12–15 additional isolates for each cereal rust fungal species. For each species these were sampled in a way to provide the best available representation of unique genotypes from global populations. This analysis demonstrated that the HD locus is multiallelic in the four species of cereal rust fungi. The four species had between 6–12 alleles of *bW-HD1* and *bE-HD2*. This supports previous analyses of the allelic diversity of these genes in *P. striiformis* f. sp. *tritici*, which identified nine alleles of *bW-HD1* and *bE-HD2* [23]. Our reported number of *bW-HD1* and *bE-HD2* alleles is likely an underestimate given the current limited sampling publicly available. Most analyzed isolates were retrieved from cereal hosts during the asexual reproduction cycle and from regions that are known to be deprived of sexually recombining populations. The only exception was *P. coronata* f. sp. *avenae* isolates, which are reported to be derived from sexual recombining populations in the United States of America [45]. We identified the highest allelic variation of *HD* genes (eleven *bW-HD1* and twelve *bE-HD2*) in *P. coronata* f. sp. *avenae*. This is similar to previous studies of isolates from likely sexual populations of *P. striiformis* f. sp. *tritici* in China, India, and Pakistan, which showed the highest diversity in *HD* alleles compared to other clonal lineages in Western wheat growing regions [23].

In contrast to multiallelic *HD* genes, the *Pra* and associated *mfa* genes were biallelic in *P. coronata* f. sp. *avenae*, *P. triticina* and *P. striiformis* f. sp. *tritici*. In all cases, one *Pra* allele was linked to one specific invariant *mfa* allele whereby *P. triticina* carried two *mfa1/3* genes that were nearly identical in the sequence of the predicted mature pheromone peptide. This biallelism of *Pra/mfa* is similar to other tetrapolar dimorphic basidiomycetes like *Microbotryum* sp., U. *maydis* and *Cryptococcus* sp. [63,90,91]. The notable exception was *P. graminis* f. sp. *tritici*, which encodes two *mfa* genes (*mfa1* and *mfa3*) in proximity to *STE3.2–3*. The two encoded precursor peptides are highly variable and are likely processed into two distinct mature

pheromone peptides with at least one amino acid difference. In addition, we cannot exclude that MFA3 encodes for additional mature pheromone peptides, which might lead to distinct receptor activation. Interestingly, *Pgt STE3.2–3* was also the most variable *Pra* allele in all species. We identified three *STE3.2–3* variants that have at least four or more non-synonymous nucleotide changes relative to the most common *Pgt STE3.2–3* variant. Moreover, the *STE3.2–3* variants in *Pgt 126–6711*, *Pgt ME-02*, and *Pgt PK-01* shared four distinctive amino acid changes in the C-terminus while being linked to an *mfa1* allele that gives rise to MFA1 with four amino acid substitutions. All analyzed *P. graminis* f. sp. *tritici* isolates carried an invariant *STE3.2–2* allele with only one *mfa* gene in proximity and we did not identify a clearly distinct additional *Pra* allele. Hence the functional significance of the observed variation at the *STE3.2–3* locus is currently difficult to assess without more extensive sampling, more phased chromosome scale genome assemblies and in the absence of direct functional crossing studies in *P. graminis* f. sp. *tritici*. Yet, the organization of the *STE3.2–3* gene is reminiscent of the PR locus of *Sporisorium reilianum* in the order Ustilaginales. In *S. reilianum*, the PR locus is at least triallelic with each allele encoding one pheromone receptor and two distinct pheromone peptides. Each of the mature pheromones derived from one allele specifically activates only one of the receptors encoded by the two other alleles [92]. More studies are needed to assess if the PR locus in *P. graminis* f. sp. *tritici* is bi- or multiallelic.

We initially predicted that biallelic MAT loci would show stronger signatures of recombination suppression and genomic degeneration than multiallelic loci based on a published mathematical model [2] and observations in other basidiomycete species [9,93]. Indeed, the multiallelic HD locus was mostly syntenic within dikaryotic genomes of the four cereal rust species. This initial analysis with a limited set of genomes per species is likely to hold true at the population level. In the case of *P. triticina*, the eight *HD* alleles of four isolates were all highly syntenic with most proximal genes conserved between the different alleles. Consistently, the composition of TEs at order classification level was highly similar between *HD* alleles of the same species.

In contrast, the PR locus showed strong signs of recombination suppression and genomic degeneration, however, the exact patterns were not identical in all four species. PR proximal regions showed the strongest signs of recombination suppression and were the most extended in *P. coronata* f. sp. *avenae*, *P. triticina* and *P. striiformis* f. sp. *tritici* with *Pra* genes located at edges or within large gene deserts and TE islands of 0.7–1.2 Mbp. At least some of these repeats were shared between the two *PR* alleles within each dikaryotic genome of *P. coronata* f. sp. *avenae* and *P. striiformis* f. sp. *tritici* based on whole chromosome alignments. This was not the case for PR alleles in *P. triticina*. In all three cases, there appeared to be differences in composition and coverage of TE at the order, superfamily, and family classification level between the two PR alleles. This was most obvious for *P. triticina*, where we analyzed four dikaryotic genome assemblies and identified a TE family specific to *STE3.2–3*, namely Ty3_Pt_*STE3.2–3*. Ty3_Pt_*STE3.2–3* displayed preferential insertion for the *STE3.2–3* allele between the pheromone receptor and the two *mfa* genes. Ty3_Pt_*STE3.2–3* is likely a currently active TE based on its sequence conservation, length of individual TE copies, and the suggested nuclear exchange between the studied Australian isolates within the last decade [37]. These initial observations of partial allele-specific TE composition and coverage in *P. coronata* f. sp. *avenae*, *P. triticina* and *P. striiformis* f. sp. *tritici* contrast with observations of TE analysis of biallelic MAT locus in 15 species of *Microbotryum*. The MAT locus in *Microbotryum* spp. did not display allele specific patterns of TE composition, yet were likely reservoirs for TE families that drove TE expansion at a genome scale at discrete time points associated with the extension of evolutionary strata around the MAT locus [42].

Compared to these observations at biallelic PR loci, the PR locus in *P. graminis* f. sp. *tritici* contained fewer TEs, more genes, and was much shorter at ~300 kbp. In addition, the TE composition and coverage was very similar between the two alleles within the dikaryotic genome of *Pgt 21–0*. These observations could be biological or a technical artifact of the assembly process of this specific genome. The *Pgt 21–0* assembly is based on older, noisy PacBio long-read technologies, included extensive manual curation and relied on gene synteny between contigs and haplotypes for scaffolding. This gene-based scaffolding might have been broken by long-stretches of TEs around the PR locus, as observed for the three other cereal rusts, and thereby introduced scaffolding errors in *Pgt 21–0* [35]. Several other genomes generated with older, error-prone PacBio long-read technologies (e. g. *Pst 104E* and *Pca 12SD80*) had issues assembling the PR locus correctly. Alternatively, these marked differences of the *P. graminis* f. sp. *tritici* PR locus might reflect biological reality, as models predict that recombination suppression and genomic degeneration is more pronounced at biallelic loci [2]. If the PR locus in *P. graminis* f. sp. *tritici* is not truly biallelic, as potentially indicated by the gene-based analysis of *Pra* and *mfa* alleles, the observed organization and level of genomic degeneration at the PR locus could support this prediction. However, more sampling and high-quality, phased chromosome-scale genome assemblies are necessary to differentiate these alternative hypotheses concerning the PR locus in *P. graminis* f. sp. *tritici*.

The functions of *MAT* genes in rust fungi are currently unknown, beyond their likely role in mediating mate compatibility. However, in *P. triticina*, *P. striiformis* f. sp. *tritici* and *M. larici-populina* [19,40,82,94], *MAT* genes are expressed during the sexual and asexual infection cycle. Suppression of *MAT* gene expression in *P. triticina* during the asexual infection cycle on wheat reduces infection severity and spore production [19]. Here, we show that *MAT* genes are also upregulated during the late stage of the asexual infection cycle of the cereal host in *P. coronata* f. sp. *avenae*, *P. graminis* f. sp. *tritici*, and *P. striiformis* f. sp. *tritici*. This suggests a broader conservation of *MAT* gene expression during the late stages of the asexual infection cycle of rust fungi. Cells of rust fungi carry more than two nuclei in the same cytosol during the asexual infection cycle [95]. Urediniospores always carry two nuclei of compatible mating-types and never two nuclei of identical mating-types. Studies from the early 20th century show that the two nuclei of urediniospore mother cells undergo synchronous nuclear division within the same cytoplasm giving rise to four nuclei within the same urediniospore precursor cells [95]. This is followed by nuclear movement to restore appropriate nuclear pairing and planar cell division that enables urediniospore maturation and reconstitutes the dikaryotic state [95]. The observed upregulation of *MAT* genes aligns with the development of urediniospore primordia and spore production. Hence, we hypothesize that *MAT* genes regulate nuclear pairing during production of dikaryotic urediniospores from primordia.

Understanding mating-types in cereal rust fungi has significant implications for agriculture and prediction of new virulent isolates. Recent reports suggest that nuclear exchange between isolates during the asexual infection cycle (also known as somatic hybridization [96]), in addition to sexual reproduction, can lead to nuclear assortment [96]. Nuclear exchange and the viability of the resulting offspring is likely regulated by *MAT* genes because all proposed events have given rise to isolates with dikaryotic genomes that carry opposing *MAT* gene pairs. We confirmed these observations with our *MAT* gene analysis in the case of *Pgt Ug99* (*Pgt 21–0* and unknown nuclear donor) and *Pt 19NSW04* (*Pt 76* and *Pt 20QLD87* as proposed nuclear donors). Conversely, our analysis showed that nuclear exchange has not occurred for the analyzed isolates of *P. coronata* f. sp. *avenae*, and *P. striiformis* f. sp. *tritici*, as none of the isolates share identical *HD* and *Pra* alleles that could be carried in the same nucleus.

In cases of nuclear exchange, these nuclei behave like entire linkage groups with virulence (also known as effectors) allele complements being tied to specific *MAT* alleles, if somatic hybridization occurs in the absence of parasexual reproduction as currently proposed. This implies that the nuclear *MAT* alleles define the possible virulence allele combinations that can arise via nuclear exchange, from either sexual or somatic reproduction. We hence can predict the most likely novel virulence allele combinations that might arise given circulating nuclear genomes in existing cereal rust fungal populations on their cereal hosts. This observation requires us to generate phased and chromosome scale genome assemblies of all cereal rust fungi at the population level. These will improve disease management strategies and the durability of characterized resistance loci in the global crop germplasm leading to smarter rust resistance breeding programs.

## Material and methods

### Assessment of used genomic resources

Detailed information on isolates and genomes used in this study can be found in Table 1. Completeness of genome assemblies were assessed by BUSCO v5.5.0 [97] in the genome mode with the basidiomycota_odb10 gene set.

### Identification of *MAT* genes

Annotation of *Pt 19NSW04* and *Pt 20QLD87* were downloaded from CSIRO Data Access Portal (URL: https://data.csiro.au/), CDS and protein data of *Pt 19NSW04* and *Pt 20QLD87* were generated with Gffread [98]. *MAT* genes including the *HD* and *Pra* genes previously identified in *P. triticina* (Table 2) [19] were used as blastp and blastn [99] query to identify orthologs in *P. graminis* f. sp. *tritici Ug99* and *21–0* [35], *P. striiformis* f. sp. *tritici 134E* [52], *104E* [53] and *DK0911* [54], *P. triticina 76* [48], *19NSW04*, *20QLD87* [37] and *15* [59], *P. polysora GD1913* [81], *P. coronate* f. sp. *avenae 203* [44], *12NC29* and *12SD80* [43] isolates. Missed *STE3.2–3* in *Pst 104E* was fixed by mapping short-read to reference genome *Pst 134E* with bwa-mem2 [100]. *Mfa genes* were identified by using *mfas* of *Pt BBBD*, *Pgt SCCL* and *Pst 78* [19] as a custom motif database to search similar pattern in all studied species with Geneious Prime v. 2022.1.1 [101], using a function built from fuzznuc [102]. Karyograms of *Pca 203*, *Pgt 21–0*, *Pt 76* and *Pst 134E* were made with RIdeogram 0.2.2 [103].

### Genealogical analysis of the *HD* and *Pra* genes

To construct a species tree for cereal rust fungi, protein sequences encoded in one nuclear genome from each cereal rust fungus species under study was used as input data. The input data was processed by Orthofinder 2.5.5 [67,104] to infer maximum likelihood trees from multiple sequence alignments (MSA). fasttree v2.1 [105] was used to calculate likelihood-based local support values. Coding regions of identified *HD* and *Pra* genes were aligned with MACSE v2.07 [106] with default setting. Coding-sequence based alignments were trimmed by using TrimAl v1.2 [107] with option (-gt 0.9) to remove sequences which have gaps in more than 10% of samples, trimmed alignments were realigned with MACSE again. Nucleotide alignments were imported into BEAUti v2.7.6 [108] to generate.xml files, bModelTest 1.3.3 [109] was used to test the best nucleotide substitution model separately with a Markov chain length of 15 million. HKY+G was chosen for inferring two *HD* genes, whereas TN93+I was chosen for *Pra* genes and *STE3.2–1* individually. Then BEAST 2 MCMC was run under a strict molecular clock with the Yule Model tree. Fifteen million length Markov chains were used to provide a sufficient effective sample size (ESS) of independent samples. Tracer v.1.7.2 [110]

was used to check if ESS values of all parameters are above 200 for each run. TreeAnnotator [108] was used to generate a maximum clade credibility tree with 10% burn in. Figtree v.1.4.4 [111] was used to visualize the final output.

Nucleotide sequences of identified *mfa* genes were aligned with MACSE v2.07 [106] with default settings, trimAl v1.2 [107] with option (-gt 0.9) to remove gaps, trimmed alignment was used to construct maximum likelihood tree via IQ-TREE 2 (version 2.1.4-beta) [112], the best model was estimated by ModelFinder (-m MFP) [113] based on Akaike Information Corrected Criterion (AICc) score [114], 10000 replicates were set for ultrafast bootstrap [115]. K2P+G4 substitution model was chosen by ModelFinder [113].

To compare topologies of *bW-HD1* and *bE-HD2* gene trees, IQ-TREE 2 (version 2.1.4-beta) [112] was used to build unrooted maximum likelihood trees for each species and both genes separately, the best model for each alignment estimated by ModelFinder (-m MFP) [113] based on AICc score [114], 10000 replicates were set for ultrafast bootstrap [115]. Top 100 optimal gene trees of *bW-HD1* and *bE-HD2* were used as alternate topologies after removing branch length values to perform AU tests [68] with IQ-TREE2 (-keep-ident–zb 10000 -n 0 -au), $p_{AU}$ <0.05 suggesting topological incongruence between the two *HD* genes.

In order to detect potential recombination events, the HD locus plus 1kb of proximal DNA sequence of each species were aligned with MAFFT v7.490 [116,117] in Geneious Prime v. 2022.1.1 [101], both sides of each alignment were trimmed manually. The RDP [118], GENE-CONV [119], BOOTSCAN [120], MAXCHI [121], CHIMAERA [122], SISCAN [123], and 3SEQ [124] methods in the program RDP5 [69] were processed to detect unique recombination events for the HD locus of each species using default settings. Recombination events which were detected by at least five of the above methods were accepted. RDP5 projects of each species are available in Dryad Database [70].

## Estimation of centromeric regions

Hi-C reads of *Pca 203* and *Pt 76* were mapped back to their corresponding haploid genome assembly with Juicer 1.6 [75] and processed with the 3d-dna [125] pipeline. Juicebox 1.8.8 [75] was used to visualize Hi-C maps. The approximate positions of centromeric regions were estimated by zooming in on the Hi-C heatmap in Juicebox and selecting the region corresponding to strong bowtie like interaction signals, which have been validated and widely used in previous studies [126, 127].

## $d_s$ value estimation for allele pairs and plotting

Inspired by the approach used by Branco et al. [8] for identifying evolutionary strata on mating-type chromosomes of *Microbotryum* species, proteinortho 6.0.22 [128] was used to pair orthologs one-to-one with -synteny and -singles tag. Muscle 3.8.1551 [129] was applied to generate coding-sequence based protein alignments with default settings. PAML 4.9 [130] was used to calculate synonymous divergence values from alignments generated with Muscle. Genomes were split in 10 kbp per window, then density of genes and repeats were calculated with bedtools v 2.29.2 [131]. Results were visualized with karyoploteR v3.17 [132].

## Synteny analysis of *MAT* genes and proximal regions

For investigating macrosyntenic relationship between haplotypes, self-alignment analyses between two haplotypes of each species were done with NUCmer (version 3.23) [72] by using–maxmatch option, alignments with a size of less than 1000 bp or less than 90% identity were removed, filtered data were visualized via Matplotlib [133].

To investigate synteny conservation, NUCmer (version 3.23) [72] was used to align nucleotide sequences of *MAT* genes and proximal regions. Blastn 2.10.0 [99] was used to identified conserved genes (identity higher than 70%, e-value < 0.05) between and within species. Results of 40 genes flanking the *MAT* genes were visualized with gggenomes 0.9.9.9000 [134].

## General repeat annotations

REPET v3.0 pipeline [135,136] was applied to annotate repeats in all genomes used in this study, together with Repbase v22.05 [137], config files were based on default settings, Wicker classification system was used to classify TE families into Class-Order-Superfamily-Family [76]. TEdenovo was used to predict novel repetitive elements and to construct specific custom databases from each genome individually as described in the manual instructions. Constructed custom databases were used for repeat annotation with TEannot. The reference TE database built for *Pt 76* from TEdenovo with the REPET v3.0 pipeline was used as reference for annotating repeats in *Pt 15* [59], *Pt 19NSW04* and *Pt 20QLD87* [37] with RepeatMasker 4.1.5 [138]. To make analyses consistent, the *Pt 76* genome was reannotated in the same way as the other three isolates.

To calculate the coverage of each TE family, classification information from TEannot was used. For TEs with more than one potential classification, hits from Repbase which have more than 70% of similarity were used as evidence for reclassification.

LTR_FINDER v1.07 [80] with ps_scan [139,140] was used to identify detailed structures of LTRs of Ty3_Pt_*STE3.2–3* in default setting. Reverse transcriptase (A0A0C4EV49_PUCT1), Retrotransposon gag domain (A0A0C4EKM8_PUCT1), Integrase, catalytic domain (A0A0C4EMG6_PUCT1) and Retropepsins (A0A0C4F2V2_PUCT1) [141], were additionally used as query to search for motifs in identified LTR retrotransposons using tblastn 2.12.0+ [99].

## Identification of variants in MAT loci

Sequence data of *Pca* were published by [45], data of *Pgt* were published by [35,46,47,58] and under PRJNA39437, data of *Pt* used in this study have been published earlier by [49–51] and under PRJNA39803 and PRJNA39801, data of *Pst* were published by [55–57] and under PRJNA60743, all above data were downloaded from NCBI SRA database (URL: https://www.ncbi.nlm.nih.gov/sra) (S1 Table). Four *Pca* isolates, *Pca 203* [44], *Pgt 21–0* [35], *Pt 76* [48] and *Pst 134E* [52], with published chromosome-scale haplotype-phased genome assemblies were used as reference for mapping. Trimmomatic 0.39.2 [142] was used to remove adapters from raw reads whereas FastQC v0.11.8 [143] was applied for assessing the quality before and after trimming. Bwa-mem2 [100] was used to map trimmed reads to reference genomes. MarkDuplicates (Picard) 1.70 [144] was applied to remove PCR-generated duplicates. Samtools 1.12 [145] was used to remove mapped reads with mapping quality lower than 30. IGV 2.16.2 [146] and Qualimap 2 [147] were both used to check mapping quality.

For reconstruction of *Pra* alleles, freebayes-parallel 1.3.6 [148,149] was applied to detect variants, bcftools 1.12 [145] was used to exclude low quality variants with -e 'QUAL<40'. Samtools 1.12 [145] was used to filter out regions of interest. The Ensembl Variant Effect Predictor (VEP) 88.9 [150] was used to annotate sequence variants. Bcftools 1.12 [145] was used to generate consensus sequences, nucleotide and protein alignments were generated and visualized with Geneious Prime v. 2022.1.1 [101].

For *de novo* reconstruction of *HD* alleles, Spades [151], a de novo assembler, was used for building draft genome assemblies from available whole-genome Illumina short-read datasets (S1 Table). For isolates with average read lengths > 101 bp, k-mer size of 101 was chosen for building draft genomes. For isolates with average read lengths < 101 bp, k-mer size of 51 bp was chosen.

*HD* genes used in genealogical analysis were used as reference genes, blastn 2.12.0+ [99] was used to search and identify *HD* allele containing contigs of the draft genome assemblies. KAT [152] was used to compute a k-mer matrix from *de novo* assembled *HD* contigs and reference *HD* loci sequences with default setting (k-mer size = 27). KAT filter was used to obtain all paired-end reads which contained *HD* k-mers based on the created *HD* k-mer matrix (-T 0.2). Spades was used to reassemble all *HD* related paired-end reads of the specific isolate. Final outputs were visualized in Geneious, CDS regions of reconstructed *HD* alleles were predicted by mapping the CDS of reference *HD* alleles to reconstructed contigs, ORFs were presumed with Geneious.

## Expression analysis of *MAT* genes

RNA-seq reads of rust fungi on infected plants were obtained from NCBI SRA database (URL: https://www.ncbi.nlm.nih.gov/sra). The datasets included: RNA-seq data of *Pst* 87/66 on wheat (PRJEB12497 [83], *Pca* 1*2NC29* on *Brachypodium distachyon* (PRJNA398546 [43], *Pgt* 21–0 on wheat (PRJNA415866 [46], *Pst 104E* on wheat (PRJNA396589 [53]. Raw reads were quality checked with FastQC v0.11.8 [143], trimmomatic 0.50 [142] was used to trim low quality reads with parameters: ILLUMINACLIP:adapter.fa:2:30:10 LEADING:3 TRAILING:3 SLIDINGWINDOW:4:15 MINLEN:36. Kallisto 0.44.0 [153] was then applied to align and count transcripts with -b 100 parameter for all paired-end reads and -b 50 -l 200 -s 20—single—single-overhang for single-end reads. *Pst 104E* cDNA data was used as reference for *Pst 104E* infected tissues. Since no reference genome of *Pst 87/66* is available, cDNA data of *Pst 134E* was used as reference instead with *HD* genes of *Pst 87/66* supplemented to obtain better quantification of the specific *HD* alleles. *Pgt 21–0* and *Pca 12NC29* were used as reference for mapping reads from *Pgt 21–0* and *Pca 12NC29* infected tissues, respectively. Gene names of *Pca 12NC29* and *Pst 104E* were added with funannotate annotate v 1.8.5 [154,155] with default parameters. Kallisto outputs were imported to EdgeR 3.17 [156] with tximportdata [157] following user instruction, followed by normalizing with the TMM method, significant upregulation of *MAT* genes was assessed with the LRT method and ggplot2 [158] was used to visualize the result. Housekeeping genes were identified in each dataset based on the following conditions: *p-value*>0.1 and log(FC) <0.5, functional annotations of candidate housekeeping genes were retrieved from interproscan [155].

## Dryad DOI

https://doi.org/10.5061/dryad.w0vt4b8zm [70].

## Supporting information

**S1 Table. List of whole-genome Sequence Read Archive data used in the present study.** The table provides information on all whole-genome Sequence Read Archive (SRA) data used in the present study. Metadata includes species abbreviation ("Species"), isolate name ("Isolate"), SRA identifier ("SRA ID"), and the initial reference ("Citation"). *Pca—P. coronata* f. sp. *avenae*, *Pgt—Puccinia graminis* f. sp. *tritici*, *Pt—P. triticina* and *Pst—P. striiformis* f. sp. *tritici*. (XLSX)

**S2 Table. List of RNAseq Sequence Read Archive data used in the present study.** The table provides information on all RNAseq Sequence Read Archive (SRA) data used in the present study. Metadata includes species abbreviation ("Species"), isolate name ("Isolate"), timepoint of infection (hpi, hours post infection) or sample type ("Sample Type"), timepoint/status of spores ("Time point/Status"), replicate number ("Replicate"), SRA identifier ("SRA ID"), and the initial reference ("Citation"). *Pca—P. coronata* f. sp. *avenae*, *Pgt—Puccinia graminis* f. sp.

*tritici*, and *Pst*—*P. striiformis* f. sp. *tritici*.
(XLSX)

**S1 Fig. Two unlinked MAT loci suggest tetrapolar mating types in four *Puccinia spp*.** Karyograms of *P. coronata* f. sp. *avenae* ("*Pca 203*"), *P. graminis* f. sp. *tritici* ("*Pgt 21–0*"), *P. triticina* ("*Pt 76*") and *P. striiformis* f. sp. *tritici* ("*Pst 134E*") with the positions of HD, PR and STE3.2–1 loci marked by black arrow heads. HD, PR and STE3.2–1, located on chromosome 4, chromosome 9 and chromosome 1, respectively, suggest tetrapolar mating types in these four species.
(TIF)

**S2 Fig. Genetic distance between pheromone precursor genes (*mfa*) and pheromone receptor (*Pra*) genes varies from 100 bp to 100 kb.** The diagrams display the location of *STE3.2–2* and *STE3.2–3* and their linked *mfa* genes on chromosome 9A and 9B in *P. coronata* f. sp. *avenae* ("*Pca 203*"), *P. graminis* f. sp. *tritici* ("*Pgt 21–0*"), *P. triticina* ("*Pt 76*") and *P. striiformis* f. sp. *tritici* ("*Pst 134E*"). The grey bar represents chromosome subsections containing the genes of interest with the numbers indicating the absolute location in mega base pairs on chromosome 9A and 9B. The genetic distance between *mfa1*/*mfa3* and *STE3.2–3* are highly variable between species, whereas *STE3.2–2* and *mfa2* are tightly linked in all species.
(TIF)

**S3 Fig. MFA protein alignments indicate mating factor a proteins are overall conserved within species.** (A) Alignment of MFA1 protein sequences from four different rust fungal species. The MFA1, which is linked to *STE3.2–3*, is mostly conserved within species. Two near identical MFA1/3 copies can be identified in four *P. triticina* isolates. In *P. graminis* f. sp. *tritici* MFA1 has four amino acid substitutions in *Pgt SCCL* versus *Pgt 21–0*. (B) Alignment of MFA2 protein sequences from four different rust fungal species. MFA2 is fully conserved within each species. (C) Alignment of MFA3 protein sequences from three *P. graminis* f. sp. *tritici* isolates. *Mfa3* is only present *P. graminis* f. sp. *tritici* in close proximity to *STE3.2–3*. Predicted mature pheromone sequences are outlined by boxes.
(TIF)

**S4 Fig. Pheromone precursor genes group according to their proximate *Pra* alleles and display strong signals of trans-specific polymorphisms.** Maximum likelihood tree of *mfas* identified in four cereal rust fungi including multiple isolates per species. Tips are labelled with the species abbreviation, gene names, and isolate names are provided in parentheses. Branch support was assessed by 10000 replicates. The scale bar represents 0.5 substitutions per site. *Pca*—*P. coronata* f. sp. *avenae*, *Pgt*—*Puccinia graminis* f. sp. *tritici*, *Pt*—*P. triticina* and *Pst*—*P. striiformis* f. sp. *tritici*.
(TIF)

**S5 Fig. Whole chromosome alignments of HD loci containing chromosomes between the two haplotypes of each dikaryotic genome exhibit slightly reduced synteny around HD locus.** The figure shows dots plots of whole chromosome alignments between the two HD loci containing chromosomes from dikaryotic genome assemblies. Each panel consists of dot plots of the whole chromosome and subset dot plot zooming into the HD locus. The HD locus is labelled and line colors show the nucleotide percentage identity and nucleotide orientation as indicated in the figure legend. Subfigure A to D show *P. coronata* f. sp. *avenae* ("*Pca 203*"), *P. graminis* f. sp. *tritici* ("*Pgt 21–0*"), *P. triticina* ("*Pt 76*") and *P. striiformis* f. sp. *tritici* ("*Pst 134E*"), respectively.
(TIF)

**S6 Fig. Allele pairs on chromosomes 4 and 9 are not overall more diverged when compared to other chromosomes.** The plots show the distribution of $d_S$ values of allele pairs on sister chromosomes in four cereal rust fungal species. Bar plots show the distribution of $d_S$ values up to the 99% quantile. Red lines represent threshold of $d_S$ values of 95% of all alleles. Blue lines represent threshold of $d_S$ values of 90% of all alleles. Black points show $d_S$ values of individual allele pairs. *HD* and *STE3* allele pairs are highlighted as red points. Subfigures A to D show *P. coronata* f. sp. *avenae* ("*Pca 203*"), *P. graminis* f. sp. *tritici* ("*Pgt 21–0*"), *P. triticina* ("*Pt 76*") and *P. striiformis* f. sp. *tritici* ("*Pst 134E*"), respectively.
(TIF)

**S7 Fig. Hi-C heatmap of *P. coronata* f. sp. *avenae* ("*Pca 203*") hapA.**
(TIF)

**S8 Fig. Hi-C heatmap of *P. coronata* f. sp. *avenae* ("*Pca 203*") hapB.**
(TIF)

**S9 Fig. Hi-C heatmap of *P. triticina* ("*Pt 76*") hapA.**
(TIF)

**S10 Fig. Hi-C heatmap of *P. triticina* ("*Pt 76*") hapB.**
(TIF)

**S11 Fig. Coverage of different transposable element orders at the HD, PR, and STE3.2–1 locus.** The plots show the percentage of nucleotides covered by different transposable element orders at (A) HD locus (B) PR locus (C) STE3.2–1 locus. Each subfigure A to C shows the coverage in each haplotype of the dikaryotic genomes of *P. coronata* f. sp. *avenae* ("*Pca 203*"), *P. graminis* f. sp. *tritici* ("*Pgt 21–0*"), *P. triticina* ("*Pt 76*") and *P. striiformis* f. sp. *tritici* ("*Pst 134E*"). Different TE orders are color coded as shown in the legend. TEs with no assigned class are labelled "Undetermined". TEs with no assigned order but belonging to Class I (RNA retrotransposons) or Class II (DNA transposons) are labelled "Undetermined Class I" or "Undetermined Class II", respectively.
(TIF)

**S12 Fig. Whole chromosome alignments of PR loci containing chromosomes between two haplotypes of each dikaryotic genome exhibit strong signs of synteny loss.** The figure shows dot plots of whole chromosome alignments between the two PR loci containing chromosomes from the dikaryotic genome assemblies. Each panel consists of dot plots of the whole chromosome and subset dot plots zooming into the PR locus. The PR locus is labelled and line colors show the nucleotide percentage identity and nucleotide orientation as indicated in the figure legend. Subfigures A to D show *P. coronata* f. sp. *avenae* ("*Pca 203*"), *P. graminis* f. sp. *tritici* ("*Pgt 21–0*"), *P. triticina* ("*Pt 76*") and *P. striiformis* f. sp. *tritici* ("*Pst 134E*"), respectively.
(TIF)

**S13 Fig. IGV screen shots of short-read Illumina mapping of various *P. coronata* f. sp. *avenae* isolates against the HD locus in the *Pca 203* reference.** (A) shows mapping against the HD locus on chromosome 4A and (B) chromosome 4B, respectively.
(TIF)

**S14 Fig. IGV screen shot of short-read Illumina mapping of various *P. graminis* f. sp. *tritici* isolates against the HD locus in the *Pgt 21–0* reference.** (A) shows mapping against the HD locus on chromosome 4A and (B) chromosome 4B, respectively.
(TIF)

**S15 Fig. IGV screen shots of short-read Illumina mapping of various *P. triticina* against the *HD* isolates locus in the *Pt 76* reference.** (A) shows mapping against the HD locus on chromosome 4A and (B) chromosome 4B, respectively.
(TIF)

**S16 Fig. IGV screen shots of short-read Illumina mapping of various *P. striiformis* f. sp. *tritici* isolates against the HD locus in the *Pst 134E* reference.** (A) shows mapping against the HD locus on chromosome 4A and (B) chromosome 4B, respectively.
(TIF)

**S17 Fig. Nucleotide alignments of the *HD* gene coding sequence in four cereal rust fungi indicate *bW-HD1* and *bE-HD2* are multiallelic in each species.** (A) Nucleotide alignment of the *de novo* reconstructed HD locus from *Pca 203* Illumina short-read data with the coding regions of *bW-HD1* and *bE-HD2* alleles from *Pca 203* dikaryotic reference genome. (B) to (D) multiple sequence alignments of *de novo* reconstructed *HD* coding regions, (E) multiple sequence alignments of *Pst bW-HD1* and *bE-HD2* alleles [159]. In each subfigure B to E, the top two track shows the consensus sequence length and relative sequence identity, respectively. Subfigure B to E show *P. coronata* f. sp. *avenae* ("*Pca*"), *P. graminis* f. sp. *tritici* ("*Pgt*"), *P. triticina* ("*Pt*") and *P. striiformis* f. sp. *tritici* ("*Pst*"), respectively. The *bW-HD1* and *bE-HD2* are numbered in accordance with Fig 1.
(TIF)

**S18 Fig. IGV screen shots of short-read Illumina mapping of various *P. coronata* f. sp. *avenae* isolates against the PR locus in the *Pca 203* reference.** (A) shows mapping against the PR locus on chromosome 9A and (B) chromosome 9B, respectively.
(TIF)

**S19 Fig. IGV screen shot of short-read Illumina mapping of various *P. graminis* f. sp. *tritici* isolates against the PR locus in the *Pgt 21–0* reference.** (A) shows mapping against the PR locus on chromosome 9A and (B) chromosome 9B, respectively.
(TIF)

**S20 Fig. IGV screen shots of short-read Illumina mapping of various *P. triticina* isolates against the PR locus in the *Pt 76* reference.** (A) shows mapping against the PR locus on chromosome 9A and (B) chromosome 9B, respectively.
(TIF)

**S21 Fig. IGV screen shots of short-read Illumina mapping of various *P. striiformis* f. sp. *tritici* isolates against the PR locus in the *Pst 134E* reference.** (A) shows mapping against the PR locus on chromosome 9A and (B) chromosome 9B, respectively.
(TIF)

**S22 Fig. Amino acid alignments of STE3.2–2 and STE3.2–3 alleles of four cereal rust fungi.** Multiple sequence alignment of *de novo* reconstructed STE3.2–2 and STE3.2–3 protein sequences. Subfigure contains MSAs; one for STE3.2–2 and one for STE3.2–3. Subfigure A to D show *P. coronata* f. sp. *avenae* ("*Pca*"), *P. graminis* f. sp. *tritici* ("*Pgt*"), *P. triticina* ("*Pt*") and *P. striiformis* f. sp. *tritici* ("*Pst*"), respectively.
(TIF)

**S23 Fig. Amino acid alignments of the three MFA alleles from various *P. graminis* f. sp. *tritici* isolates.** Amino acid substitutions are highlighted by color, whereas predicted mature pheromone sequences are outlined by boxes.
(TIF)

**S24 Fig. Whole chromosome alignments of *HD* genes containing chromosomes between haplotypes of four different *P. triticina* isolates.** The figure shows dots plots of whole chromosome alignments of HD loci containing chromosomes derived from distinct dikaryotic genomes of four different *P. triticina* isolates including *Pt 15*, *Pt 19NSW04*, *Pt 20QLD87* against *Pt 76*. Each panel consists of a dot plot of the whole chromosome and a subset dot plot zooming into the HD locus. The HD locus is labelled and line colors show the nucleotide percentage identity and nucleotide orientation as indicated in the figure legend. (A) Comparison of nucleotide sequence of chromosome 4s of *Pt 15* and *Pt 76*. (B) Comparison of nucleotide sequence of chromosome 4s of *Pt 19NSW04* and *Pt 76*. (C) Comparison of nucleotide sequence of chromosome 4s of *Pt 20QLD87* and *Pt 76*. (D) Comparison of nucleotide sequence of chromosome 4s of *Pt 19NSW04* and *Pt 20QLD87*.
(TIF)

**S25 Fig. The HD locus is highly conserved in four *P. triticina* isolates.** Synteny graphs of *HD* loci including proximal regions in the four *P. triticina* isolates *Pt 15*, *Pt 19NSW04*, *Pt 20QLD87* and *Pt 76*. Red lines between chromosome sections represent gene pairs with nucleotide sequence identity higher than 70% and grey shades between conserved nucleotide sequences ($>= 1000$ bp and identity $>= 90\%$). For additional annotations please refer to the provide legend ("Legend").
(TIF)

**S26 Fig. Coverage of different transposable element orders at the HD locus in four *P. triticina* isolates.** The plots show the percentage of nucleotides covered by different transposable element orders at the HD locus. Each subfigure shows the coverage in each haplotype of the dikaryotic genomes of *P. triticina* isolates *Pt 15*, *Pt 19NSW04*, *Pt 20QLD87* and *Pt 76*. Different TE orders are color coded as shown in the legend. TEs with no assigned class are labelled "Undetermined". TEs with no assigned order but belonging to Class I (RNA retrotransposons) or Class II (DNA transposons) are labelled "Undetermined Class I" or "Undetermined Class II", respectively.
(TIF)

**S27 Fig. Whole chromosome alignments of *Pra* gene containing chromosomes between haplotypes of four different *P. triticina* isolates.** The figure shows dot plots of whole chromosome alignments of *STE3.2–2* or *STE3.2–3* containing chromosomes derived from distinct dikaryotic genomes of four different *P. triticina* isolates including *Pt 15*, *Pt 19NSW04*, *Pt 20QLD87* against *Pt 76*. Each panel consists of a dot plot of the whole chromosome and a subset dot plot zooming into the proximal region of *STE3.2–2* or *STE3.2–3*. The *STE3.2–2* or *STE3.2-* are labelled and line colors show the nucleotide percentage identity and nucleotide orientation as indicated in the figure legend. (A) Comparison of nucleotide sequence of chromosome 9s containing *STE3.2–2* gene (B) Comparison of nucleotide sequence of chromosome 9s containing *STE3.2–3* gene.
(TIF)

**S28 Fig. Nucleotide coverage distribution of Ty3_Pt_*STE3.2–3* on the chromosomes of four *P. triticina* isolates.** The plots show the percentage of nucleotides covered by the transposable element family Ty3_Pt_*STE3.2–3* on each chromosome of four *P. triticina* isolates *Pt 15* (A), *Pt 19NSW04* (B), *Pt 20QLD87* (C) and *Pt 76* (D). Chromosomes carrying *STE3.2–3* are highlighted in red.
(TIF)

**S29 Fig. Schematic illustration of the location of Ty3_Pt_*STE3.2–3* copies at the *STE3.2–3* locus in four *P. triticina* isolates.**
(TIF)

**S30 Fig. Genealogy of *STE3.2–1* in rust fungi indicates sequence of *STE3.2–1* are conserved within species.** Bayesian rooted gene tree built from *STE3.2–1* coding-based sequence alignment from four cereal rust fungi: *P. coronata* f. sp. *avenae* (*Pca*), *P. graminis* f. sp. *tritici* (*Pgt*), *P. triticina* (*Pt*) and *P. striiformis* f. sp. *tritici* (*Pst*) and *P. polysora* f. sp. *zeae* (*Ppz*) GD1913 was included as outgroup. Trees are based on a TN93+I model of molecular evolution. Each node is labelled with its values of posterior probability (PP). PP values above 0.95 are considered to have strong evidence for monophyly of a clade and PP values of identical alleles are not displayed. The scale bar represents the number of nucleotide substitutions per site. Alleles of the same species are colored with identical background: *Pca* (yellow), *Pgt* (green), *Pt* (blue), *Pst* (orange).
(TIF)

**S31 Fig. Whole chromosome alignments of *STE3.2–1* gene containing chromosomes between two haplotypes of each dikaryotic genome suggest high conservation of *STE3.2–1*.** The figure shows dot plots of whole chromosome alignments between the two *STE3.2–1* gene containing chromosomes from dikaryotic genome assemblies. Each panel consists of dot plots of the whole chromosome and subset dot plots zooming into the *STE3.2–1* proximal region. The position of *STE3.2–1* gene is labelled, and line colors show the nucleotide percentage identity and nucleotide orientation as indicated in the figure legend. Subfigures A to D show *P. coronata* f. sp. *avenae* ("*Pca 203*"), *P. graminis* f. sp. *tritici* ("*Pgt 21–0*"), *P. triticina* ("*Pt 76*") and *P. striiformis* f. sp. *tritici* ("*Pst 134E*"), respectively.
(TIF)

**S32 Fig. Synteny analysis of *STE3.2–1* genes and their flanking regions reveal no evidence of genetic degeneration.** Synteny graphs of the *STE3.2–1* locus including proximal regions in *P. coronara* f. sp. *avenae* ("*Pca 203*"), *P. gramminis* f. sp. *tritici* ("*Pgt 21–0*"), *P. triticina* ("*Pt 76*"), and *P. striiformis* f. sp. *tritici* ("*Pst 134E*"). Proximal regions are defined as 40 genes downstream and upstream of the *STE3.2–1* alleles, respectively. *STE3.2–1* proximal regions are highly syntenic within each dikaryotic genome and conserved between species. Red lines between chromosome sections represent gene pairs with sequence identity higher than 70% and grey shades represent conserved nucleotide sequences ($> = 1000$ bp and identity $> = 90\%$). For additional annotations please refer to the provided legend ("Legend").
(TIF)

**S33 Fig. Synonymous divergence ($d_S$) values between *STE3.2–1* alleles along chromosome 1A suggest conservation of this locus.** Synonymous divergence values ($d_S$) for all allele pairs are plotted along chromosome 1A for (A) *P. coronara* f. sp. *avenae* ("*Pca 203*"), (B) *P. gramminis* f. sp. *tritici* ("*Pgt 21–0*"), (C) *P. triticina* ("*Pt 76*"), and (D) *P. striiformis* f. sp. *tritici* ("*Pst 134E*"). In each panel, the top track shows the $d_S$ values ("$d_S$") of allele pairs along chromosome 1. Each dot corresponds to the $d_S$ value of a single allele pair. The second and third track show the averaged TE ("TE") and gene ("gene") density along chromosome 1 in 10 kbp-sized windows, respectively. The *STE3.2–1* alleles are highlighted with a red line and red shading indicates a 0.4 mbp-sized window around the *STE3.2–1* genes. Predicted centromeric regions are marked with blue shading. The two lower tracks ($d_S$ values and gene locations) provide a detailed zoomed in view of red shaded area around the *STE3.2–1* alleles. Species-specific background coloring is the same as for Fig 1.
(TIF)

**S34 Fig. Multidimensional scaling (MDS) plot of RNA dataset used in this study.** MDS plots were made with TMM normalized counts for quality control, each dot represents a single sample, and replicates were color coded as indicated in each subfigure legend. (A) MDS plot of TMM-normalized value of *Pca* at 48 hour post infection (hpi) and 120 hpi. (B) MDS plot of TMM-normalized value of *Pgt* at 48, 72, 96, 120, 148 and 168 hpi. (C) MDS plot of *Pst* at 24, 48, 72, 120, 168, 216 and 264 hpi. (D) MDS plot of TMM-normalized value of ungerminated spore (US), germinated spores (GS) stages, 144 hpi, 216 hpi and haustoria enriched samples (HE) of *Pst*.
(TIF)

**S35 Fig. Expression of housekeeping genes in four RNAseq datasets.** (A) TMM-normalized value of housekeeping genes in *P. coronata* f. sp. *avenae* ("*Pca 12NC29*"), *P. graminis* f. sp. *tritici* ("*Pgt 21–0*"), and *P. striiformis* f. sp. *tritici* ("*Pst 87/66*" and "*Pst 104E*"). (B) Likelihood ratio test (LRT) method was applied to test significant upregulation of housekeeping genes between timepoints, none of the housekeeping genes show significant upregulation between timepoints. red dashed lines indicate logFC = 0.5 and logFC = -0.5 respectively, stars above or below bars indicate statistically significant differences between the two adjacent time points: *$p<0.05$, **$p<0.01$, ***$p<0.001$. Genes are labelled with different colors. Transcription elongation factor TFIIS (*TFIIS*), Actin-related protein 3 (*ARP3*), Actin/actin-like protein 2 (*ACTIN2*), Ubiquitin carboxyl-terminal hydrolase 6 (*UBP6*), Conserved oligomeric Golgi complex subunit 3 (*COG3*), Ctr copper transporter 2 (*CTR2*).
(TIF)

**S36 Fig. Upregulation of *MAT* genes in late stages of the asexual life cycle.** Likelihood ratio test (LRT) method was applied to test significant upregulation of *MAT* genes between timepoints, red dashed lines indicate logFC = 0.5 and logFC = -0.5 respectively, stars above or below bars indicate statistically significant differences between the two adjacent time points: *$p<0.05$, **$p<0.01$, ***$p<0.001$. *MAT* genes were labelled with different colors. (A) The expression levels of *MAT* genes in *P. coronata* f. sp. *avenae* ("*Pca 12NC29*") were compared between 120 hours post infection (hpi) and 48 hpi. (B) The expression levels of *MAT* genes in *P. graminis* f. sp. *tritici* ("*Pgt 21–0*") were compared between 72 hpi and 48 hpi, 96 hpi and 72 hpi, 120 hpi and 96 hpi, 144 hpi and 120 hpi, 168 hpi and 144 hpi. (C) The expression levels of *MAT* genes in *P. striiformis* f. sp. *tritici* ("*Pst 87/66*") were compared between 48 hpi and 24 hpi, 72 hpi and 48 hpi, 120 hpi and 72 hpi, 168 hpi and 120 hpi, 216 hpi and 168 hpi, 264 hpi and 216 hpi. (D) The expression levels of *MAT* genes in *P. striiformis* f. sp. *tritici* ("*Pst 104E*") were compared between germinated spores (GS) and ungerminated spores (US), 144 hpi and GS, 216 hpi and 144 hpi, haustoria enriched samples (HE) and 216 hpi.
(TIF)

**S37 Fig. *MAT* genes are upregulated in the late asexual infection stage of *P. striiformis* f. sp. *tritici* ("*Pst 104E*").** TMM-normalized value of *MAT* genes in ungerminated spores (US), germinated spores (GS) stages, 144 hpi, 216 hpi and in haustoria enriched samples (HE) of *Pst*. Genes are labelled with different colours.
(TIF)

# Acknowledgments

We thank Dr. P. Tobias and Dr. M. Moeller kindly for critical reading and comments on this manuscript. We thank Dr. M. Bui for advice in iqtree2 usage, Prof. C. Linde for useful

suggestions, and R. Tam for providing a snakemake pipeline for trimming raw sequencing data and centromere information of *Pst 134E*. We thank J. Lin for suggestions on data visualization. This work was supported by computational resources provided by the Australian Government through the National Computational Infrastructure (NCI) under the ANU Merit Allocation Scheme.

## Author Contributions

**Conceptualization:** Benjamin Schwessinger.

**Data curation:** Zhenyan Luo.

**Formal analysis:** Zhenyan Luo, Alistair McTaggart, Benjamin Schwessinger.

**Funding acquisition:** Benjamin Schwessinger.

**Investigation:** Zhenyan Luo.

**Methodology:** Zhenyan Luo, Benjamin Schwessinger.

**Project administration:** Benjamin Schwessinger.

**Resources:** Benjamin Schwessinger.

**Software:** Zhenyan Luo, Benjamin Schwessinger.

**Supervision:** Benjamin Schwessinger.

**Validation:** Zhenyan Luo.

**Visualization:** Zhenyan Luo, Alistair McTaggart.

**Writing – original draft:** Zhenyan Luo, Alistair McTaggart, Benjamin Schwessinger.

**Writing – review & editing:** Zhenyan Luo, Alistair McTaggart, Benjamin Schwessinger.

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
