## [Decision Letter · Decision Letter 0]

22 Sep 2023

Dear Dr Schwessinger,

Thank you very much for submitting your Research Article entitled 'Genome biology and evolution of mating type loci in four cereal rust fungi' to PLOS Genetics.

The manuscript was fully evaluated at the editorial level and by independent peer reviewers. The reviewers appreciated the attention to an important problem, but raised some substantial concerns about the current manuscript. Based on the reviews, we will not be able to accept this version of the manuscript, but we would be willing to review a much-revised version. We cannot, of course, promise publication at that time.

If you decide to revise the manuscript for further consideration at PLOS Genetics, please aim to resubmit within the next 60 days, unless it will take extra time to address the concerns of the reviewers, in which case we would appreciate an expected resubmission date by email to plosgenetics@plos.org.

We are sorry that we cannot be more positive about your manuscript at this stage. Please do not hesitate to contact us if you have any concerns or questions.

Yours sincerely,

Tatiana Giraud

Guest Editor

PLOS Genetics

Eva Stukenbrock

Section Editor

PLOS Genetics

The manuscript has been evaluated by three referees, who agree that this manuscript on the mating-type loci of rust fungi could be of interest for PLoS Genetics, providing valuable insights into mating-type loci in rust fungi. Despite the positive assets acknowledged by the referees, however, serious concerns were raised, in particular about methods, generalization from one isolate to species, unwarranted conclusions about HD gene recombination, language, format, clarity of the text and figures, lack of citations of previous works, the inclusion of a single PR allele in gene genealogies, which does not allow analysing trans-specific polymorphism, lack of discussion on the drivers of recombination suppression, lack of information on the size of recombination suppression, lack of analyses of polymorphism data and of recombination suppression at HD genes, and lack of data availability. I have to agree with these concerns. The manuscript is not very well written, many sentences are unclear or make no sense, are awkward or with wrong syntax. The introduction includes irrelevant information about resistance genes but lack essential ones, and the predictions at the end of the introduction seem ad hoc, while more interesting one could be stated. Some examples are highlighted below and by referees but the text needs to be edited all along the manuscript, and the figures improved. I would encourage resubmission only if you are able to revise the manuscript along these lines and add the corresponding analyses. The referees also provide a list of excellent additional suggestions and questions, which should also be addressed. Please also be sure to cite previous works on mating-type loci in rusts.

More specific suggestions

-Please number the pages

-L29: It's not the loci that give rise to recombination suppression, they undergo recombination suppression; It's not necessarily the linkage either.

-L60: I would rather say "sex and mating-type”

-L66-68, L516 : it’s not really « initial sheltering » but rather « a combination of an initial advantage of carrying fewer deleterious mutations than average and the sheltering of these few deleterious mutations when increasing in frequency ». I would also highlight here the relevant prediction made in Jay et al 2022 that recombination suppression extension should only occur in dikaryotic species, and only around biallelic loci not multi-allelic ones. It would thus fits the findings that it occurs around the biallelic PR locus and not the multiallelic HD locus, and come back to this in the discussion.

-L90, L93 : advantageOUS

-L93 : awkward sentence. It’s not that « multi allelism » is advantagous (this is wrong group selection reasonning), but that any new, rare allele is advantageous, which leads to multi-allelism ; this reasonnning is very different.

-L98 : awkward sentence : the mating system is selfing versus outcrossing while the mating compatibility system is bipolar or tetrapolar, this is very different and the relationships between the two systems is not simple. A tetrapolar compatibility system does not decrease inbreeding (you can always self), but it is less advantageous under inbreeding, this is very different. I would delete this whole paragraph, not relevant to the study anyway.

-L105-132 : delete also this paragraph, it’s not relevant to the study and there are several awkward sentences (for example an isolate is not « adapted », it’s populations that adapt not individuals). Just give the minimal information on the role of MAT genes in the life cycle.

-L133 : why « likely » ? based on what ?

-L136 : MAT loci not alleles. See https://www.nature.com/articles/s41467-023-41413-4.pdf

-L143-150 : awkward predictions, we don’t understand why, they seem like ad hoc, after you knew the results. I would rather explain predictions about recombination suppression around biallelic loci and not multiallelic ones as explained above based on Jay et al. 2022.

-avoid abbreviations, and in particular stick to conventional Latin abbreviations for species names.

-Table legends are too short, remove capitals, explain abbreviations, explain all lines and columns

-L184-186 : awkward sentence

-L201 : « fully conserved » is unclear : present ? at the same location ? identical sequence ?

-L205 : « composition » is unclear : identical sequence ?

-L218 : see Petit et al (2012) Evolution 66: 3519–3533.

-L222, L252, L522 : explain better what trans-specific polymorphism means and refer to the first paper on this on the PR gene : Devier et al 2009 Genetics 181:209-223. Predictions about this in the introduction would be better than the predictions currently at the end of the introduction.

-L232 : either rather than both

-L235 : unclear what is your initial hypothesis and why. This whole page is hard to follow, having the last sentence at the beginning would help (but see referees’ comments on this inference, likely unreliable due to low bootstraps)

-L261 : run a formal test of congruence

-L267 : these are not phylogenetic trees but gene genealogies (phylogenies are for species)

-L271 : alleles and not genes

-L300 HD locus

-L310 and elsewhere : S in dS should be subscript

-L326 : delete « pair » and « are »

-L397 : conservation is unclear : in synteny ? sequence ? within instead of « of each »

-L398 : any conserved

-L404 : awkward wording

-L401 : ancestral to what ?

-L415 : awkward wording

-L428 : no plural at polymorphism

-L439-440 : this makes no sense.

-L445-449 : unclear ; what is the « status » ? conserved means identical sequences ?

-L453 : I don’t think this is true beyond this particular species in the given reference

-L457 : delete significant

-L460 : describe is not the right term here

-L463, L466 : we’re lost with the jargon here, these terms should be explained in the introdcution instead of the long irrelevant paragraph on R genes. Explain whether these findings are consistent with exectations given the life cycle and where the MAT genes act.

-L482 : mating compatibility not sexual reproduction, this is very different

-L487 : make it clear whether or not other genes are trapped into the non-recombining region

-L489 : the PR locus

-L491 : the hypothesis

-L492 : delete parasexual reproduction, this has not been defined nor shown.

-L494 : mating is not tetrapolar, mating compatibility is, this is very different

-L506 : why mitosis activity ?

-L508 : delete « between haplotypes but » and begin a new sentence with « Chromosomes… »

-L519 : likely caused ; See https://www.nature.com/articles/s41467-023-41413-4.pdf

-L528 : awkward sentence, it’s more balancing selection

-L529 : awkward sentence, the causal « because « sounds odd : it’s not the cause but the observation allowing the inference

-L532 : see referees’ concerns about this, I agree this should not work and is poorly supported.

-L537 : it’s not consistent but similar

-L545-6 : this is wrong group selection reasonning : selection does not work for the good of the species

-L548 : pheromone receptor gene and singular or locus ?

-550 : at each ?

-L556 : unclear

-L558-560 : unclear and unconvincing, delete

-L566 : unclear ; of different mating types ?

-L573-3 : this should be explained in the introduction instead of the long digression about R genes

-L576 : unclear ; is a verb missing ?

-L582-596 : this makes no sense and is irelevant, delete

-L601 : demete between species, pleonasm

Reviewer's Responses to Questions

**Comments to the Authors:**

Reviewer #1: In this study, the group of B. Schwessinger made use of several Puccinia striiformis f.sp tritici (Pst) genome assemblies which they generated previously; they were among the first pioneering the proper haplophasing of the two haploid genomes residing in the two haploid nuclei in each urediniospore (the source of gDNA used for sequencing projects). Here, they compared these to genomes from three other Puccinia species pathogenic to cereals (mostly wheat), and whose haplophased genomes were recently published. They present the organization and variation of the MAT loci (HD and PRA/mfa) among these species/ forma speciales. Though the various mating-type genes (HD and PRA/mfa) have been reported on elsewhere in a few studies including their likely tetrapolar arrangement in Puccinia species, there has not yet been done a more in depth comparison of their arrangement and genomic neighborhood in various Puccinia species (the four presented here). This is becoming feasible only recently because of the above-mentioned haplophased assemblies. From the comparisons, they infer evolutionary principles of the MAT loci for these rust fungi.

Claims:

- the tetrapolar arrangement of the HD and PRA/mfa mating-type loci in all four species; has been substantiated

- similar, little degeneration around HD loci is found when comparing haplotype regions, as to other genomic regions as an indication of low recombination suppression; has been substantiated

- PRA/mfa mating-type locus much more degenerated among haplotypes, surrounded by higher levels of repetitive DNA (transposable elements or TEs, and low gene density / “gene desserts”) as other genome regions, as an indication of recombination suppression preceding speciation; substantiated to a certain degree but see comments below on TEs

- tried to locate centromers and link these to mat loci; see comments below

- the HD genes are multiallelic “highly polymorphic” within various species isolates; see comments below

- the PRA locus is biallelic with one allele in each haplotype (STE3.2-2 and STE3.2-3) in the sampled species; substantiated. Potential polymorphisms may indicate more allelic variants, particularly in Pgt (correlating with potential three allelic PRA genes?); see comments below

- use of publicly available transcriptome data for the four species indicates both MAT loci are expressed during the asexual infection cycle and higher expression is seen in later stages of the infection (very few data exists on expression during the sexual stages on alternate hosts); tentative, see comments below.

Overall, a nice assessment of MAT loci in this group of fungi, but some issues would need to be resolved before statements / claims can be made.

Line 350-351: “Composition of transposons vary across species, DNA transposons dominated the HD locus of Pst134E but HD loci of other rust fungi had higher coverage of RNA transposons (S9 Fig).”

And Line 400-401: “Pst 210 and Pt 76 showed strong signals of accumulation of independent transposable element families in each nuclear haplotype, which indicates invasion by TEs since cereal rusts shared a most recent common ancestor.”

- As the supplementary figure only contains single representatives of the species, can species-wide conclusions be drawn? Could you provide more explanation on the families of TEs and how the results were generated for figure S9?

Lines 568-575: “detailed transcriptomic analysis of MAT genes now gives an initial indication that these genes are important for nuclear pairing in the asexual stage, which is consistent with earlier preliminary reports in Pt (21) and M. larici-populina (62, 63). Our analysis suggests that MAT genes are expressed in asexual life cycle stages of cereal rust fungi. In our detailed transcriptomic analysis, we found both HD and PRA genes were upregulated during the infection of the asexual host with a peak at final infection stages that align with spore production and dispersal HD genes are known to control post-mating growth, such as control development of fruiting-body in some Ascomycota and Basidiomycota species (64).“ Also Figure 6.

- Does the up-regulation of these genes really warrant this conclusion? Moreover, from figure 6C, bE2-HD2 does not change in expression. Based on the plot, between 24h and 48h, there is a lot of variation for the bW1-HD1, bW2-HD1, and Ste3.2-3 transcripts, and using the median point no increase of expression when compared to 5days post infection, is there an explanation?

- Is there a lot of variation between genes at the early time points? For early time points, is there is a possibility that there is not enough fungal biomass to get meaningful number of reads, or depth, to make these inferences?

- Is it possible to get information from kallisto or downstream EdgeR of how many genes have counts at the early time points relative to spore or late sample times. And do you have sufficient counts at these early time points for differential expression analysis? Another quick method would be to look at housekeeping genes throughout time points or genes that don’t change.

- Also, there is / we could not find any reference to parameters used with EdgeR within the github, only deseq2, the parameters and methodology would be good to see.

Issues with Methods.

Lines 631 -635: “Estimation of centromeric regions…. Hi-C reads of Pca 203 and Pt 76 were mapped back to corresponding diploid genome assembly with Juicer 1.6 (71) and processed with 3d-dna (72) pipeline. Juicerbox 1.8.8 (71) was used to visualize Hi-C maps and manually estimate centromeric regions by identifying strong interaction signals.”

- Please, include the Hi-C contact maps used for centromeric identification. As discussed with respect to the PR locus, this region is marked by repeats and transposable elements which would affect the mapping of Hi-C reads, how can you be confident where the centromere is? Is there a centromeric specific sequence motif between the analyzed isolates like found in other fungi?

- I would if it was mentioned at the beginning of the methods that specific scripts and parameters used in the study are in the attached git repo. Also, include how REPET/TEdenovo was done in the main methods, as transposable elements are in the main figures.

Lines 658-664: “…before and after trimming. Bwa-mem2 (65) was used to map trimmed reads to reference genome, MarkDuplicates (Picard) 1.70 (92) was applied to remove PCR-generated duplicates. Qualimap 2 (93) was used to check mapping coverage and quality. Freebayes-parallel 1.3.6 (94, 95) was processed to detect variants, samtools 1.12 (96) was used to filter out regions of interest. The Ensembl Variant Effect Predictor (VEP) 88.9 (97) was used to annotate sequence variants. Bcftools 1.12 (96) was used to generate consensus sequence, nucleotide and protein alignments were generated and visualized by Geneious Prime v. 2022.1.1 (66).”

- Could you provide coverage plots for each of the genes for all the isolates to ensure that the genes are fully covered by short reads, and that there is sufficient coverage for consensus sequences to be created from the variants.

- Are there regions where reads are not aligning due to differences? This has also implications for the (HD) allele diversity you claim.

Other issues and suggestions:

Line 76: …’ to govern nuclear compatibility and inbreeding within populations’; maybe better: ‘… to govern mate selection within populations and nuclear compatibility’? It is not solely for “inbreeding”.

Line 84: maybe add ‘..which are outwardly transcribed in opposite directions.’?

Line 85: ‘cascade’, one word

Line 87: ‘…and the involvement of one or both loci in mating’?? Better: ‘Different segregation patterns of these mating-type loci define the terms bipolar or tetrapolar mate compatibility.’

Line Suggested: ‘…are physically linked and hence only one MAT locus controls mate compatibility.’

Line 90: advantageous

Line 98: The tetrapolar..

Line 101-104: A very nice new example is: Coelho et al., 2023: doi:10.1073/pnas.2305094120

Line 113: ‘is prevented by management strategies that remove the sexual host e.g. Berberis’. I think, though this has helped in the past, the fact that the alternate host(s) may not be available in that region of production of the primary host, may be more important?

Line 118: ‘it poses serious risks’

Line 122: Suggestion: ‘As alternatives to sexual reproduction and gene assorting, somatic hybridisation in which nuclei are exchanged during hyphal interactions can increase genotypic diversity including the generation of novel effector complements (29, 30).’

Line 126: ref 31 is cited but this publication is based on a seriously flawed Pt genome assembly (this publication should be retracted and not cited). If Pt genome assembly information is needed, a better resource is a preprint by Sperschneider et al. bioRxiv preprint https://doi.org/10.1101/2022.11.28.518271

Line 146: delete ‘hypothesis’

Line 157 – 159: “Here we made use of the four available chromosome-level genome assemblies of cereal rust fungi at the time of the study including Pca 203, Pst 134E, Pt 76 and Pgt 210 (Table 1). We included P. polysora f. sp. zeae (40) or Melampsora larici-populina as an outgroup.”

- It would be nice to see in the table which assemblies are complete/assembly information sizes etc.

Line 169: ‘…that are thought to bind to’; this has not yet been proven, and is thought to be analogous to other, more-studied basidiomycete fungi….

Line 174 – 175: “In the case of Pst 104E, STE3.2-2 was absent from the genome assembly but we recovered its haplotype by mapping raw sequencing reads of Pst 104E to the Pst 134E phased chromosome scale genome assembly (34)”.

- Could you clarify what was done here? Did you map reads then assemble this region ? Or take the consensus of the mapped reads as STE3.2-2. ?

Line 216: ‘…is also diverse…’

Line 392: ‘the PRA genes of Pst 134E were found to be ~164kbp from the centromere of chromosome 9A and ~473kbp from that of chromosome 9B, likely physically-linked to their centromeres’.

- to which figure is this referring, and are these typos? Those distances don’t look like “physically-linked’?!

Line 317-318: “The information of centromere locations suggests that the HD loci are likely not directly linked to centromeres in all four species”.

- This was based on juicer hi-c contact plots, can they be included in the supplementary?

Line 386-387: “Gene density dropped significantly around PRA genes with STE3.2-2/STE3.2-3 located within or adjacent to ‘gene deserts’ that extended over 1mbps in the case of Pst 134E.”

- Could this be a factor of current annotation limits as these regions contain large repetitive regions / transposons and normally these would be masked in conventional annotation pipelines? Do all gene annotations have RNA support, or vice versa, are there RNA reads mapping to areas where no genes have been “annotated”?

Line 423: I do not see the Pca HD maps in this S14 figure to substantiate the text….?

Line 424-430: “This is likely caused by the fact that isolates selected were mostly linked to sexual populations of Pca found in the USA (49). Hence we also used the most closely related available reference of each sample for short read mapping purposes including references from Pca 203 (50), Pca 12NC29 and Pca 12SD80 (49). This showed that the HD loci in Pca display high levels of polymorphisms including heterozygous SNPs, multiple unaligned regions and gaps, which suggests the HD loci present in our selected Pca isolates are not well reflected in the available references.”

- Would you expect this for the other isolates?

Line 439-444: “We identified slightly more SNPs in STE3.2-2 and STE3.2-3 of Pt which we estimate to give rise to a maximum of four distinct haplotypes. Yet each predicted haplotype has only a very limited number of amino acid substitutions. Pgt was the most polymorphic at the STE3.2-2 and STE3.2-3 loci. We identified three haplotypes of STE3.2-2 with several amino acid changes. STE3.2-3 was the most polymorphic in Pgt including two isolates, TTTSK and UVPgt60, which contain potential nonsense mutations leading to pre-mature stop codons.”

- If these regions are marked by increase of repeats, what is the mapability of the short reads to these genes and loci?

And, as mentioned at line 513-514: “and complete loss of synteny within and among species surrounding he PR loci”.

- Doesn't this suggest that there will be issues of mapping from other isolates to these regions if haplotypes of the same isolate show high heterozygosity for the PR loci as in Fig 4 B?

Line 484: ‘two loci that control mate compatibility’; I think such statements (and in general throughout the manuscript, e.g. also lines 548-550) need to be presented as tentative since no functional molecular work has demonstrated this in the rusts. All based on analogous work in other basidiomycetes, primarily Ustilago and Cryptococcus.

Line 485: “…HD loci are highly multiallelic”; again, overstated since only cursory, indirect data from a few isolates per species has been presented; see comment above on potential problems with mapability of (Illumina) short reads.

Line 519: “…likely be caused’; ‘…is likely caused’

Lines 520-521: “Interestingly, the TE families surrounding PR loci are not shared within or among species.”

- Again, is this a valid claim when only looking at four species and single isolates from each?

Line 555 and on, Discussion on potential multiallelic PRA alleles in Pgt; there is precedence in Sporisorium reilianum: Schirawski et al. 2005 DOI:10.1128/EC.4.8.1317-1327.2005

Line 564 – 566: “This is even though nuclei of the same mating type share the same cytoplasm during colonisation of the asexual host."

- I do not understand this comment. If it read “…sexual host...”, it would maybe make sense ((haploid) teliospores undergo another mitotic division so a dikaryotic cell / infection hypha directly penetrates the alternate host leaf / epidermal cell.

Line 567: …is unknown; I don’t think that this preliminary transcript analysis is enough to prove that ‘… (MAT genes) are important for nuclear pairing in the asexual stage’. Some experimentation (Cuomo et al., 2017) suggests that they affect overall virulence…..

Line 574: ‘…and dispersal.’ Period

Lines 582-586: “Individual nuclear genomes of rust fungi are entire linkage groups, with effector haplotype complements being tied to specific MAT haplotypes if somatic hybridisation occurs in the absence of parasexual reproduction. Hence, we can predict the possible combinations of effector complements that can arise by somatic hybridisation based on the knowledge to which MAT haplotypes they are linked to.”

- Doesn’t this suggest that there are only a few combinations based on haplotypes that can exist? Can this claim be made when there is a large amount of transposable elements? Are you confident there is no asexual mechanism for information exchange between haplotypes during somatic hybridization?

Fig 1. Pca203: bW26HD1? (delete 6?); Pca12SD80 should be bW6-HD1?? It’s confusing. Please, call haplotype allele numbers consistently. Check all!

Fig 2. Wrt incongruency and inconsistency in a phylogenetic tree when analyzing STE3.2-2 and STE3.2-3 together; in several analyses in other studies, the pheromone receptor sequences were C-terminally truncated to exclude the cytoplasmic tail to optimize the alignment and allow various PRA alleles to be directly compared (e.g., Coelho et al. 2017, Cuomo et al. 2017; Bakkeren et al., 2008,FGB 45,S15). No detail is given on how the pile-ups and analyses were done.

Legend Fig S10: ‘yellow boxes indicate PR loci’; I do not see yellow boxes….?

Reviewer #2: Luo and colleagues offer a genome-level analysis of mating type (MAT) loci in four economically important cereal rust species. The study reveals that these species possess a tetrapolar mating system characterized by unlinked biallelic pheromone/receptor (PR) locus and a multiallelic homeodomain transcription factor (HD) locus. Through a range of genomic analyses, the authors report signs of recombination suppression and genomic degeneration around the PR locus. Conversely, the HD locus appears to be more conserved and does not show signs of genetic degeneration, suggesting ongoing recombination in its neighboring regions as normally observed in other tetrapolar basidiomycetes. The work also emphasizes the potential role of these MAT loci in promoting nuclear pairing during the asexual cycle, offering insights into the evolution of virulence in these critical pathogens. Overall, the paper has the potential to significantly advance our understanding of mating compatibility mechanisms in this important group of fungi with possible repercussions for agriculture and disease management.

While the manuscript provides valuable insights into mating type loci in rust fungi, it needs several revisions. First, the language needs polishing for clarity and readability, and typographical errors and inconsistent formatting should be fixed. Second, the figures need a thorough review to improve readability and presentation. Finally, the Methods section is too vague, making it hard to understand the research process and raising concerns about reproducibility. These general issues should be addressed alongside my more specific comments/suggestions below.

- Major points -

Line 83: It appears that the manuscript uses reversed nomenclature for the bW (bWest) and bE (bEast) genes, potentially following the precedent set in a previous paper on rust fungi mating type loci (Cuomo et al. 2017, G3, 7:361-376). However, based on foundational research on mating type loci in smut fungi (Ustilaginomycotina), the bW gene corresponds to HD2 and bE to HD1. Therefore, I recommend revising this information for the sake of accuracy and ensuring that this corrected nomenclature is consistently applied throughout the manuscript, so that it can be adopted and reflected in future studies.

Lines 163-164: The manuscript establishes that the HD and PR loci are located on separate chromosomes, thereby supporting a tetrapolar MAT organization. Given the significance of this finding to the overall narrative of the study, it would be beneficial to emphasize this information through visual representation in a main figure. Specifically, I recommend the addition of a figure (potentially as Fig. 1) that provides a simplified version of what is currently shown in S1 Fig. This figure could graphically depict the chromosomes containing the PR and HD loci for each of the four studied species. To provide additional evolutionary context, this could be juxtaposed with the species phylogeny that is currently presented in S4 Fig. Merging these key elements into a main figure would enhance the comparative aspects of the study and facilitate a more immediate understanding of the findings.

Lines 205-207: The authors present interesting findings regarding the genomic organization of pheromone receptor and pheromone genes across the Puccinia species examined. Specifically, they note that STE3.2-2 is found adjacent to MFA2, while MFA1 is located at varying distances from STE3.2-3. Additionally, in Pt, two identical MFA1 genes were identified, possibly indicative of a recent duplication event. On the other hand, in Pgt, two distinct pheromone genes (MFA1 and MFA3) were found in association with STE3.2-3. These observations raise several questions: (1) Why were the two identical genes in Pt labeled as MFA1/3? Is this a nomenclatural choice or does it imply something about the gene function? (2) Given that MFA3 is unique to Pgt, it could be that this gene product may not function in mating type recognition, or might not even be a mating pheromone? (3) Could the authors speculate on what might constitute the protein sequence of the mature, active form of the pheromone? (4) How does this compare with other Pucciniales previously sequenced (e.g., Melampsora larici-populina).

Lines 234-241: The authors suggest that the differing branching patterns of HD1 and HD2 within each species may indicate recombination between these genes. This could potentially lead to self-compatible HD1-HD2 allele pairs. However, an alternative explanation could be a lack of strong phylogenetic signal, as evidenced by some low bootstrap values. I recommend a more cautious interpretation of these results. Additionally, a phylogenetic analysis using protein sequences might be informative. It's important to see if the nucleotide differences result in functionally distinct proteins. Also, in many other basidiomycetes, the N-terminal regions of HD1 and HD2 are more variable, possibly affecting self/non-self-recognition. Have you explored this in the available HD alleles?

Lines 263-264: The conclusion in this section suggests that the incongruent topologies of STE3.2-2 and STE3.2-3 indicate long-term recombination suppression around the PR locus. However, the actual evidence supporting long-term recombination suppression would be phylogenetic clustering of alleles by mating type across species, rather than clustering by species (i.e., trans-specific polymorphism). This is because the two alleles ceased recombining long ago and have subsequently accumulated substitutions independently. It is also worth noting that the incongruent topologies between the two trees are associated with relatively low bootstrap values in the STE3.2-2 tree. Again, these low bootstrap values may affect the confidence level of the inferred phylogenetic relationships, potentially complicating the interpretation.

Lines 450-478: This study provides important data on mating type gene expression. Since STE3.2-1 doesn't appear to be mating type-specific based on your results, it might be useful to examine if this gene is upregulated under the conditions you studied, or how its expression varies during infection or spore formation. This could give more context on a possible role of this gene. Additionally, you might consider referencing similar genes in other basidiomycetes that don't directly affect mating type. For instance, in Cryptococcus deneoformans, the Cpr2 gene competes with the Ste3 receptor and triggers unisexual reproduction when overexpressed. Similarly, in some mushroom species, receptor-like genes near the PR locus are linked to vegetative growth, not mating type (e.g., Wirth S, et al. 2021). This could hint at broader functions for genes like STE3.2-1 in rust fungi.

- Minor points -

Line 43: The term “di-om mating” appears in the abstract but is not explained or mentioned elsewhere in the manuscript. The terminology is not only unclear but might also be too technical for a broader readership at the abstract level. Did you intend to write “di-mon mating”?

Lines 62-65: The sentence “Genetic degeneration footprints are higher rates of (non)synonymous substitutions (dN, dS), accumulation of transposable elements, reduced gene expression and reduced gene numbers which are all a consequence of recombination cessation” could benefit from clarification. Are all these “footprints” of genetic degeneration always observed in non-recombining regions?

Line 72: Please consider changing “100s” to “hundreds”.

Line 74: According to standard taxonomic nomenclature, the abbreviation “sp.” in Microbotryum sp. should not be italicized. Furthermore, if referring to multiple species within the Microbotryum genus, the correct form should be “Microbotryum spp.” Please verify this and other instances throughout the text for consistency, such as on lines 101 and 501.

Line 77: The phrase “In most cases, two MAT loci control mating” could be more accurately stated as, “In most cases, two MAT loci determine mating type identity.”

Lines 78-79: To enhance clarity and consistency in the manuscript, consider standardizing the nomenclature for the pheromone receptor genes, choosing either “PRA” or “STE3” for uniform reference throughout. Additionally, please take the opportunity to review and standardize the formatting of gene names (italics vs. non-italics, uppercase vs. lowercase) in both the main text and figure legends, such as in S2 Fig.

Line 81: The term “PR haplotype” is used, which may not align well with conventional nomenclature for individual variations of a gene or locus, especially in the context of mating type loci literature. To ensure clarity and consistency, consider using the term “PR allele” or “idiomorph” instead. Furthermore, when discussing STE3.2-2 and STE3.2-3, it might be more appropriate to refer to these as two different alleles of the same gene. This would align better with terminology used in other parts of the manuscript, such as lines 252-253, and contribute to a more cohesive narrative."

Line 84: It is important to clarify that the products of HD1 and HD2 from the same allele do not form heterodimers. Heterodimerization occurs exclusively between HD1 and HD2 products that are derived from different alleles. Consequently, these products must differ in compatible mating partners.

Line 90: The term “Biopolar” should be corrected to “Bipolar,” and the phrase “are advantages in” should be revised to “are advantageous in.” Similar corrections are needed on line 93.

Line 91: The phrase “that undergo selfings” should be corrected to “that undergo selfing.”

Lines 93-95: The sentence, “In outcrossing populations it is advantages that MAT loci are multiallelic because this increases the success of compatible gametes (in doing what?) derived from two independent individuals,” needs revision for clarity and completeness.

Lines 93-95: The sentence regarding the advantages of multiallelic MAT loci in outcrossing populations is currently unclear and incomplete. Specifically, it would be helpful to clarify what is meant by “increases the success of compatible gametes.” Are you referring to a higher likelihood of successful fertilization, greater genetic diversity, or some other form of “success”?

Lines 96-97: The segment “with ten to hundred known and estimated haplotypes” is unclear and needs clarification.

Line 98: While the term “plesiomorphic” is technically accurate, it might come across as overly technical and potentially less accessible to a broader audience. In this case, it might be beneficial to use the term “ancestral” as a replacement to enhance readability.

Lines 100-103: To offer more background and improve clarity, consider rewording this section along the lines of: “For example, Microbotryum spp. are highly selfing, and multiple independent events have linked the HD locus and the PR locus into large non-recombining regions. These regions differ in both their size and age and have been formed by successive steps of recombination suppression, resulting in several adjacent “evolutionary strata” of differentiation between non-recombination chromosomes.

Line 112-113: You may want to rephrase the sentence to: “as sexual reproduction is prevented by management strategies that remove the secondary host (e.g., Berberis) required for sexual reproduction.”

Lines 116-117: To better articulate the point, consider revising the sentence to: “The asexual epidemic potential of a given rust isolate is largely determined by the complement of effector genes encoded in its two nuclei, as well as by the resistance (R) genes present in crop varieties grown in a specific region, which confer them the ability to recognize and mount defenses against specific pathogens or their effectors”.

Lines 146: “loci” should not be italicized.

Line 158: There seems to be a discrepancy in the notation for “Pgt 210.” Is this the same as “Ptg 21-0”? If so, please consider standardizing.

Line 184: For enhanced clarity, consider rewording the sentence along the lines of: “We employed a phylogenetic approach to trace the inheritance of PRA and HD genes in each haplotype. Under a tetrapolar system, compatible haplotypes in a dikaryotic genotype are expected to have one copy of either STE3.2-2 or STE3.2-3 and exhibit different alleles at the HD locus.”

Lines 207-208: To conform with standard notation, please separate the numbers from the units in the manuscript. For example, change “0.54Mb” to “0.54 Mb” and “10kb” to “10 kb,” and apply this consistently throughout the text.

Line 222: For the mention of trans-specific polymorphism, consider adding a relevant citation to provide further context and support for this concept.

Line 308: The use of the term “sister chromosomes” could introduce ambiguity. I recommend using the term “homologous chromosomes” instead. Additionally, it would be insightful if the manuscript could also address the extent of synteny conservation among the MAT chromosomes across the species studied. Providing information on the average level of divergence between the analyzed haplotypes in different species would also be beneficial.

Line 314-318: The phrase “user features” in this sentence is unclear and might cause confusion. Could you please elaborate on what you mean by this term or consider using a more straightforward expression? Also, it is mentioned that no “obviously aberrant patterns of gene or transposon density” were observed around the HD locus. Could you specify whether there is a particular analysis that supports this assertion? Lastly, I believe the methods used to predict the centromere locations are not very well discussed – if these were determined based on previous research, please provide the appropriate citations.

Lines 386-387: It is noteworthy that in Pca and Pst, there is a marked drop in gene density on only one side of the pheromone receptor gene locus. Could this asymmetry be directly correlated with the position of the pheromone genes that is distinct between the two PR alleles/idiomorphs?

Line 423: It is stated that “Pca showed the highest level of variation of HD loci at the species level (S14 Fig).” However, upon reviewing S14 Fig, I noticed that this figure only displays HD1-HD2 allele variation for three of the species (Pst, Pgt, and Pt). Therefore, the data supporting this claim for Pca appears to be missing or not clearly presented in the supplemental figure. Could you please clarify or correct this?”

Line 439: A higher number of SNPs in the STE3.2-2 and STE3.2-3 genes was found in Pgt compared to other species. Could this increase be attributed to the Pgt isolates analyzed be more divergent? It might be helpful to expand on this by comparing SNP frequencies in other genomic regions as well. For instance, S16 Fig. suggests that STE3.2-1, which is not directly related to mating type determination based on your findings, also seems to exhibit a high level of variation. This could indicate that the elevated SNP levels in STE3.2-2 and STE3.2-3 may not be gene-specific but rather a broader characteristic of the Pgt isolates examined? If so, this might challenge the suggestion made in Discussion section (lines 489-490) that “Pgt might be multiallelic at the PR locus.”

Lines 503-507: About the role of STE3.2-1, I find it important that it should be mentioned that other non-mating type pheromone-like receptor genes have been found in other basidiomycetes, including the Cpr2, which is known in Cryptococcus deneoformans to compete with the Ste3 receptor for signaling and whose overexpression elicits unisexual reproduction (Hsueh et al., 2009, EMBO J. 28:1220-1233). In mushroom species, pheromone receptor-like genes are also found close to the PR locus, but these do not determine mating type identity. A recent report suggest they may have a role in vegetative growth, and self-signaling (e.g., see Wirth S, et al. 2021, J Fungi, 7:399), thus revealing an important role for hyphal growth and development.

Lines 526-528: While it is appropriately highlighted the presence of trans-specific polymorphism in the STE3 alleles of rust fungi, it might be beneficial to the discussion to also note that STE3.1 and STE3.2 alleles in other Pucciniomycotina species such as Microbotryum, as well as in red yeasts and Leucosporidiales (for further context, see Coelho et al. 2010, PLoS Genetics 6:e1001052 and Maia et al., 2015, Genetics 201:75-89), appear to be evolutionarily older than those specific to the Pucciniales. This contextual information could provide additional depth to your discussion on the evolutionary aspects of these alleles.

Line 539: Typo in the term “Pucciniomycotina”

Lines 600-601: As it reads, the statement "There is no evidence of recombination suppression at HD loci, as there are no trans-specific polymorphisms between species and support for macro-synteny among species" appears to be contradictory or unclear.

Line 625: As stated above, the Method section requires careful revision and added details. For example, in the “Phylogenetic analysis of the HD and PRA genes” subsection, the absence of parameters used in OrthoFinder for orthology inference and in IQ-TREE2 for phylogenetic analysis is a notable omission.

Line 637: Should be Branco et al.

Fig 3, Line 329: The phrase "10k bp sized windows" should be amended to "10 kb-sized windows". Moreover, the function of the shaded black/grey areas below the dS plots is unclear. If these are intended to represent gene-containing regions, it would be helpful to make this explicit in the figure legend or main text. I also noticed that not all shaded areas have corresponding dS values. If this discrepancy exists because an allele is missing in one of the two haplotypes, please clarify that point. My concerns extend to the calculation of dS values, as mentioned in my comment for S8 Fig below. Additionally, to improve visual distinction between the tracks, consider applying different color schemes for Transposable Elements (TEs) and genes. Lastly, as a general comment for all figures in the manuscript, maintaining a consistent color code for labeling different species would enhance readability and interpretation, as observed in Figures 1 and 2.

Fig 4: The gene tracks are rather small, making it difficult distinguishing individual genes. To enhance readability, I suggest enlarging the gene tracks.

S2 Fig. Line 36: Please correct the phrase “cross species” to “across species” for clarity.

S3 Fig. Line 47: The word “there” appears to be a typographical error and should be corrected to “their.”

S4 Fig: To aid in cross-referencing between the text and the figure, consider adding the species abbreviations next to each species name in the phylogenetic tree.

S5 Fig: I suggest that the phylogenetic tree be rooted at the midpoint. This would prevent inadvertently implying a directional bias in the tree when using the MlpPh1 gene as the outgroup.

S8 Fig: The figure includes calculated dS values for the STE3.2-2 and STE3.2-3 genes (also plotted in Fig 5). Given that these two genes are not easily alignable, it would be helpful to clarify the methodology used to calculate these dS values. Additionally, for enhanced transparency, consider providing the percentage of pairs of genes for which dS was possible to be calculated in each chromosome across all studied species and/or include the associated raw data of these dS calculations.

References Section: I noticed several inconsistencies in the formatting of the references, particularly concerning the use of italics. For example, references 33 and 59 do not follow the same formatting as others in the list. Please carefully revise the entire section.

Reviewer #3: In this study the author investigated the structure, evolution and function of mating type loci in several cereal rust fungi using publicly available dataset of genomes and expression data analysis.

Very little is known on the evolution of mating type loci in rusts compared to other Basidiomycota and their study can be highly relevant to better understand sexual and asexual cycle of these damaging pathogens.

The study goal is very interesting and I didn’t detect any major methodological flaws. However, I found overall that the state of art was not enough described, some part of the material and method were not enough detailed and some results need to be better discussed in their context.

Major comments:

-For example, recombination suppression around the PR locus was shown previously to be likely ancestral to Basidiomycota (see Devier et al 2008 Genetics); this doesn’t appear clearly in neither the abstract, the introduction and the discussion of the results.

-The estimated size of the region with recombination suppression found around PR locus is not clearly described; how much this region varied in size between species ? between haplotypes?

-The gene tree for the PR locus should include both haplotypes to conclude to transpecific polymorphism (Fig2). Transpecific polymorphism is clear from mfa genes however.

-It is to note that there seems to have some haplotype clustering within species for bW-HD1 gene both for Pt and Pst species. This may suggest some recent recombination suppression (Fig1), which is consistent with high dS value for this gene.

-Looking at the allelic diversity within species, I wonder if the authors consider retrieving the PR/HD sequences by blast analysis in de novo assemblies; read mapping in a region rich in TE as in the PR region may be challenging. The method of estimation of the number of alleles at HD and PR locus need to be further described

-the finding that the MAT genes are expressed late in the asexual infection cycle during spore production is interesting ; however, it is likely not the only gene to be overexpressed in this stage; more evidence / functional analysis need to be done to check that MAT loci govern compatibility of nuclei in somatic hybridization; L576-582: conclusions need to be alleviated

-the evolutionary drivers of recombination suppressions are presented in the introduction, but the finding of recombination suppression around the PR loci is not discussed in this context in the discussion.

Minor comments:

Introduction

-l60-68: other models predict recombination suppression around mating-type loci and could be presented here

-l75: more Basidiomycota examples could be quoted as recombination suppression was described in some species of Cryptococcus, Agaricus, Ustilago… It could be interesting to describe if recombination suppression was associated to bipolarity/tetrapolarity, mating system, number of alleles at mating-type loci in these systems.

-l93: typo for “advantages”

-l101: the recurrent comparison with Microbotryum should be justified with a phylogenetic context that may not be obvious to readers; furthermore evolution of recombination suppression in this genus should be better descrivzs: if it is in bipolar or tetrapolar species, around HD or PR genes or linking them, etc…

-l105-106: this sentence is not related to the previous paragraph and the main goal of the study in my opinion

-l143-145: More context need to be given to understand why the author have the presented

hypothesrs ; I failed to understand why the authors tested these hypotheses in particular when reading the introduction.

Results

- Table 1: statistics about the genome assemblies (N90, L50 …) seems lacking and should be presented

Author mention “chromosome scale” assemblies, are telomere and centromere well identified? Is it haplotype-phased assemblies? Are all genomes dikaryotic genotypes?

-l184: “phylogenetic hypothesis” is not clear

-l345: previous sentences state the opposite

-l393: a distance of 473 kb to the centromere may be high for physical linkage; are other elements supporting linkage to the centromere such as segregation analyses?

Material & method

-l621: the mapping approach needs to be futher described; how was the sequence retrieved after mapping?

-l623: the search of orthologs with GeniousPrime needs to be further described; which dataset was used?

-l626-627: which dataset was used to build the rooted species tree? Is it coding sequence?

-L637: typo “Branco et al”

-One separate section could summarize all strains/genomic data used in this study

-the RNAseq dataset used (time points, number of replicates by time points) need to be better described

-TE annotation and gene models used need further description

**Have all data underlying the figures and results presented in the manuscript been provided?**

Reviewer #1: **No: **See comments to the authors. For example, more data on TE categories; better numerical data on gene transcript levels / read depths (especially wrt HD allele variants calling); parameters used with EdgeR within the github; Hi-C contact maps / data; identification of centromers.

Reviewer #2: **No: **For example, the absence of raw data for both the dS calculations and the RNA-seq analyses is noticeable. Including these as supplementary datasets would enhance the transparency and reproducibility of the work.

Reviewer #3: Yes

PLOS authors have the option to publish the peer review history of their article (what does this mean?). If published, this will include your full peer review and any attached files.

Reviewer #1: No

Reviewer #2: No

Reviewer #3: No

---

## [Decision Letter · Decision Letter 1]

29 Jan 2024

Dear Dr Schwessinger,

Thank you very much for submitting your Research Article entitled 'Genome biology and evolution of mating type loci in four cereal rust fungi' to PLOS Genetics.

The manuscript was fully evaluated at the editorial level and by independent peer reviewers. The reviewers appreciated the attention to an important topic but identified some concerns that we ask you address in a revised manuscript.

We therefore ask you to modify the manuscript according to the review recommendations. Your revisions should address the specific points made by each reviewer.

Yours sincerely,

Tatiana Giraud

Guest Editor

PLOS Genetics

Eva Stukenbrock

Section Editor

PLOS Genetics

The revised manuscript has been evaluated by the same three referees, who agree that this manuscript on the mating-type loci of rust fungi has been improved following their suggestions. The referees nevertheless still had extensive concerns, in particular about the surprising similarity between PR alleles, the delimitation of the mating-type loci, the clustering of transposable elements as well as the lack of clarity and length of the text.

I have to agree with these concerns. About the clustering of Ty elements, note that a previous study found such enrichment in non-recombining regions of mating-type chromosomes (https://doi.org/10.1038/s41467-023-41413-4). In addition, the manuscript is still not very well written. It should really be corrected by a native English specialist in the field. Some issues are highlighted below, but there are many others all along the manuscript.

I would encourage resubmission only if you are able to revise the manuscript along these lines. The referees also provide a list of excellent additional suggestions and questions, which should also be addressed.

More specific suggestions:

-L66: it’s not alleviated but increased!

-L34, L564: it’s not the mating types that are tetrapolar but the species.

-L40, L578: correlation is between two quantitative variables; what is an allele status? Allele number?

-L42: what does evolutionary conservation mean here?

-L71-76: awkward and not really exact. A correct formulation would be for example: “… suggest that non-recombining fragments could fix more easily when linked to biallelic permanently heterozygous loci such as mating-type loci due to the sheltering of deleterious mutations. Indeed, non-recombining fragments with fewer deleterious mutations than average are beneficial and can rise in frequency, but they will suffer from exposing their (few) recessive deleterious mutations at a homozygous stage when becoming frequent, except if they are linked to a permanently heterozygous locus that will shelter them.”

-L77: predicts that

-L85-95 : keep the general ideas first (i.e. from L91) and the specific statements later (i.e. from L85).

-L107, L118 : « heterothallic » is not a mating system (heterothallic fungi can undergo both outcrossing and diploid selfing)

-L211 : tetrapolar is not a mating system either, it does not control selfing and outcrossing either

-L221 : suggested that (« that » should not be omitted in written text)

-L370 : consistent with what ? what does « order level » mean ?

-L374, L404, 462 : what is conserved ? gene order ? sequenced identiry ? « conserved » makes no sense « within genomes », conserved applies to long evolutionary times, do you mean are similar ?

-L382, 383 and elsewhere : why loci plural ? There is a single locus, even across species, these are not different loci

-L388 : why mbps plural ?

-l389 : well, it’s not concomittant, it’s causal : it’s because TE have accumulated that the regions are gene poor

-L422 : awkward : the polymorphism exist regardless of whether you map (so not only « when mapping « )

-L426 : isolates are not populations

-L430 : again, the locus is the same across species so the formulation is incorrect, the locus is not specific to Pca, the sequence is

-L449 : gave rise is not optimal formulation

-All along the ms : check that you add a hyphen between two words that qualifies a third one, e.g. L477 protein-coding gene, or L455 chromosome-scale, otherwise your long sentences are hard to read

P21-22 : this is hard to read, much too detailed

-L519 : MAT loci are not involved in sexual reproduction per se but in mate compatibility, this is very different. The HD locus is also involed in dikaryotic growth, why shouldn’t this be also the case in asexual reproduction ?

-L541 : any expression

-L565 : as already said in a previous version, it’s not that multiallelism is beneficial under outcrossing (this is wrong group selection reasonning, evolution does not act for the good of populatio,), but that rare alleles are selection for under outcrossing and leads to multi-allelism, this is very different.

-L624 : they actually follow the same trend

Reviewer's Responses to Questions

**Comments to the Authors:**

Reviewer #1: Comments on rebuttal (Jan 2024)

I very much appreciate that the authors took most of the comments and issues from the 4 (!) reviewers to heart and seriously revamped the study. Hence, in my opinion, the manuscript has been substantially improved. All of the issues we raised have been addressed, which made the supplementary data set quite large: but the many added mapping data, alignments and redone (molecular) phylogenies made the data much more supportive of the conclusions. Also, generating the many HD alleles de novo for proper comparisons and subsequent conclusions added much clarity to the manuscript.

The intro is now much more focused on the evolution of the mating-type region and known literature; many concepts are now introduced.

Some minor suggestions:

Line 25, right at the start of the abstract: I don’t quite get the term “Obligate heterozygous loci such as”. I would delete this whole first sentence since you come back later to it.

Line 27, delete “for”

Line 28: suggestion: “To date, an analysis of genome-level mating-type (MAT) loci is lacking in the obligate biotrophic basidiomycetes in….”

Line 32: “P. graminis….”

Line 34: “tetrapolar mating systems in the Pucciniales. The HD locus is found to be multiallelic….”

Line 36: “were..”; switching to past tense compared to he sentence above. Be consistent!

Line 39: “…with a clear biallelic PR locus”; you already said this a few sentences before….

Line 43: “…is related to correct nuclear pairing during spore formation.”; is unclear. IS meant: “..may play a role in maintenance of a proper dinuclear state.”, or something?

Line 85: delete “behind”

Line 143: “The role of MAT genes during the life cycle of rust fungi is rudimentary. Though it is hypothesized that MAT genes regulate appropriate nuclear paring during dikaryotic spore production and mediate the compatibility of haploid cell fusions on the “alternate” host”; I do not understand the first sentence, what is meant by “rudimentary”, the knowledge of their role(s)? Also, the hypothesis brought forward is heavily based on molecular work in the smuts; maybe this should be mentioned (and a ref).

Line 202: “unequivocal”; in the abstract the term “appears to be biallelic” is used, which is more carefully phrased. You show the presence of two alleles here in the few surveyed isolates; it’s nuanced in the Discussion, with Pgt potentially have more alleles. This may may be confirmed when more isolates genomes are generated worldwide in the future. Even for the other three investigates species; can this be excluded?

Line 292: I am not sure that the two genes bE-HD2 and bW-HD1 actually share the same promotor; yes, they are tightly linked, physically close together and outwardly expressed and may depend on similar/the same transcription induction elements, but I don’t think the can be defined as the same promoter. Experiments will need to be done to ascertain this.

Line 296: “…. , which might be caused by recombination within the HD locus.” Could different rates of genetic drift (accumulation of nucleotide mutations) followed by different selection pressures be the cause?

Line 662. Wrt MAT genes expression during spore stages, Zhan e al., 2023: https://doi.org/10.1007/s44154-023-00107-z, can be cited here again (you already mentioned this ref earlier). Ironically, their data set, though publicly available, cannot easily be checked for these MAT genes!

Data availability: the link to the repository Dryad (https://doi.org/10.5061/dryad.w0vt4b8zm) does not seem to work.

Reviewer #2: Luo and colleagues have commendably addressed the concerns and suggestions from my initial review, significantly enhancing the quality and clarity of their manuscript. Notably, improvements in terminology, formatting, figures, and content have been made. The extension of the methods section, coupled with the provision of raw data on Dryad and analytic code on GitHub, is particularly appreciated for its contribution to transparency and reproducibility.

In response to specific points raised in my initial review, the authors have opted to maintain the nomenclature of HD1 (as bWest) and HD2 (as bEast) genes, but with added clarifications provided in the text. Additionally, they provide a more comprehensive approach to explore HD allele diversity and of variability in the N-terminal regions of HD. The authors also acknowledge the limits of their dataset regarding STE3.2-1 expression and added relevant references to similar findings in other fungi.

In this revised evaluation, I have focused on assessing how these and other changes have impacted the overall narrative and quality of the manuscript. Below, I provide additional comments and suggestions on specific points. Overall, the manuscript has undergone significant improvements and is moving closer to aligning with the high standards of PLOS Genetics.

Major points

1) PR and HD locus regions: The manuscript would benefit greatly from a clearer delineation of what the authors consider to be the PR locus region in each of the haplotypes for the species studied. While the dot plot analysis provides valuable information, an overarching view of the synteny for the PR-containing chromosomes in each species would significantly enhance the clarity and comprehensiveness of the findings. The HD locus region (only HD1-HD2?) should also be highlighted.

2) Mature Pheromone Peptides Predictions: In the section concerning the predicted Mfa1 and Mfa3 proteins in Pgt (Lines 266-268 and Fig S3), the authors mention the possibility of distinct mature pheromone peptides being produced by these two genes, each potentially having different receptor specificities. However, the authors don’t provide the predicted sequences for these mature peptides. Clarifying these sequences is important for assessing their specificity and understanding the functional implications of these differences. Upon closer examination, it appears that the Mfa1 precursor may encode multiple copies of the putative peptide moiety. For instance, the Ptmfa1/3 precursor contains two copies of the sequence “QWGNGSHIC” with CAAX motifs separated by a spacer sequence. This contrasts with the Mfa2 precursors, which seem to encode only one mature peptide. This pattern of multiple peptide encoding is also observed in other Pucciniomycotina species, such as Rhodotorula spp. and Microbotryum spp. and also noted in other Pucciniales.

3) Hi-C data and PR/HD locus linkage to centromeres: The conclusion that HD loci are likely not directly linked to centromeres, derived from Hi-C data, warrants further clarification. How does this method definitively determine the absence of direct (genetic?) linkage, and were these assessments also made for the PR locus? The potential limitations in resolution or mapping in complex genomic regions should also be addressed.

4) Evolutionary history of STE3.2-2 and STE3.2-3 alleles: An intriguing aspect noted is the considerable identity shared between the STE3.2-2 and STE3.2-3 alleles within the same species (~43% identity, ~56% similarity). This raises an interesting question about their evolutionary history. Do you think this level of similarity could indicate a more recent allele turnover, potentially following the loss of the STE3.1 allele variant? This seems particularly noteworthy when considering that in other Pucciniomycotina species, the STE3.1 and STE3.2 alleles are more divergent and often not alignable for dS calculations. Furthermore, I am curious about the divergence rates of non-MAT genes within the presumed PR locus region. Specifically, do these non-MAT genes exhibit increased synonymous substitution rates (dS) compared to other genes located on the same chromosome but outside the putative PR region? This assumes, of course, that there are genes common to both haplotypes within the PR region, allowing for such a comparison. Alternatively, is the gene content within the PR region too divergent between the haplotypes to enable a meaningful comparison of substitution rates?

5) Clustering of Ty3_Pt_STE3.2-3 TE family: The observation that the Ty3_Pt_STE3.2-3 TE family predominantly clusters on chromosomes containing the STE3.2-3 locus, particularly within a region bordered by the pheromone and receptor genes, is a notable finding. This specific clustering suggests restriction of this element in this specific genomic region. I’m curious about several aspects regarding this observation. Firstly, what might be the potential mechanisms that restrict the Ty3_Pt_STE3.2-3 to the PR STE3.2-3 locus? Could there be unique genomic or epigenetic features in the vicinity of the STE3.2-3 locus that might explain why this TE family does not transpose more frequently to other genomic regions? Secondly, Have similar specific TE elements been observed within the STE3.2-2-containing haplotype? Thirdly, the absence of a GAG domain in Ty3_Pt_STE3.2-3 TEs is intriguing. Given that GAG domains are typically associated with the structural proteins of TEs, could this absence affect the mobility or functionality of these TEs?

Minor points:

1) Lines 28-30, Abstract section: The statement, "To date, genome-level mating type (MAT) loci analysis is lacking for obligate biotrophic basidiomycetes in the order Pucciniales, which contains many economically important plant pathogens," could be refined to highlight the novel aspects of the study more explicitly, by utilizing chromosome-level and phased genome assemblies to analyze mating type (MAT) loci.

2) Line 95/97: “PRA” and “MFA” nomenclature. Please double-check and confirm that the use of this nomenclature for proteins and genes is the adopted by the journal and fungal genetics.

3) Line 97: use of “matching allelic PR locus”, perhaps would more clearly expressed as “compatible PR allele.”

4) Line 100: The phrase “also known as bE-HD1 and bW-HD2” could be rephrased for clarity, and to provide a historical context, to "originally designated as bE-HD1 and bW-HD2 in Ustilago maydis” and include in addition, the original reference where this designation was first established.

5) Line 121: “ten to hundred known…”, consider revise to “ten to hundreds of known…”

6) Lines 140-143: The discussion about cell types and ploidy changes during the life cycle is important in the context of understanding the evolution of non-recombining regions. Therefore, it might be beneficial to consider citing and contextualizing this information in light of the recent findings by Talhinhas et al. (Microbiol Spectr. 2023 11:e0153223. doi: 10.1128/spectrum.01532-23). This study reports the widespread occurrence of replicating haploid and diploid nuclei in various life cycle stages of Pucciniales species, suggesting a unique life cycle that is distinct from traditional haplontic, diplontic, or haplodiplontic cycles.

7) Lines 293-295: In the context of discussing tree topologies in phylogenetic analysis, using the term "congruent" instead of "concurrent" might be more accurate and descriptive. Furthermore, since the authors suggest the possibility of recombination within the HD locus, conducting a more direct test of recombination employing software like RDP5 could help substantiate this hypothesis. Although different from the hypothesis proposed here, I also recommend considering the study conducted on Ustilago maydis as a pertinent reference as it demonstrates how novel HD specificities can arise from single homologous recombination events, leading to simultaneous changes in the dimerization subdomains of the bE and bW proteins (see Kämper J, et al. 2020. New Phytol. 228:1001-1010. doi: 10.1111/nph.16755; also check Perlin MH. 2020. New Phytol. 228:799-801. doi: 10.1111/nph.16847).

8) Lines 387-391: In the section discussing gene density around PRA genes, it is mentioned that STE3.2-2/STE3.2-3 are located within or adjacent to 'gene deserts' and these regions are enriched in TEs. However, the text refers to Pgt 21-0 as an exception, but it's not entirely clear what this exception pertains to. Does the exception refer to the lack of a 'gene desert', the absence of TE enrichment, or a different pattern of gene density around the PR locus in Pgt 21-0? Clarification on this point would help in better understanding the unique characteristics of the PR locus in Pgt 21-0 compared to the other species studied.

9) Upon reviewing the data presented in S12 Fig related to the TE composition in Pra proximate regions, I noticed that although the proportions of different transposable elements vary, it appears that the classes of TEs present are more or less the same across the different cereal rust fungi. This observation seems somewhat contradictory to the statement in the manuscript (lines 399-402) about the varied TE composition and accumulation in these regions. If so, please clarify.

10) Line 421: S18 Fig seems to refer to Pgt not Pca.

11) Line 434: Not sure that “de novo assembly” should be considered a “novel” approach.

12) Line 455-461: Regarding the assessment of more than two functional Pra alleles in Pgt, I am not entirely convinced by this conclusion. It seems more likely that the observed variations are variants of the same allele class, a common occurrence when comparing more divergent strains or lineages. A more cautious or nuanced discussion of this finding would be beneficial, considering the high degree of allele variation in fungal species. Additionally, the authors indicate that mfa1 and mfa3 in Pgt had several non synonymous changes and that this is consistent with significant variation observed in STE3.2-2 and STE3.2-3. However, I think would be informative to determine if the observed variations at the mfa1 and mfa3 loci translate into differences in the mature peptides produced.

13) Lines 498-506: In the section detailing TE analysis around the STE3.2-3 and STE3.2-2 loci, I have some points and suggestions to enhance clarity and comparability of the presented data. First, the current format of presenting Fig 6C and S30B separately makes direct comparison challenging. I suggest either placing these figures together or, alternatively, generating a plot that highlights the TE ratio or differences between STE3.2-3 and STE3.2-2. Secondly, the 'empty' portions of the bars in Fig 6C related to LTRs are not immediately clear and should be explained. Third, regarding the quantification of LTR Ty3 TE Family in Fig 6C, the text mentions that Pt 20QLD87 has the highest coverage of the LTR Ty3 TE family (Ty3_Pt_STE3.2-3), as depicted in Fig 6D. However, it's not evident from Fig 6C how this conclusion is drawn, as the coverage of this specific TE family doesn’t seem to be separately quantified in that figure. Could you elucidate how this conclusion was reached or consider revising Fig 6C to explicitly demonstrate the coverage of Ty3_Pt_STE3.2-3?

14) I have noticed that the supplementary figures in the provided PDF are of generally low resolution. It's possible that this issue might have arisen due to the PDF conversion process during submission, but please double-check.

Reviewer #3: In this study the author investigated the structure, evolution and function of mating type loci in several rust fungi using publicly available dataset of genomes and expression data analysis.

I found that overall the authors addressed many comments of the reviewers on the previously submitted version and that this version of the manuscript is very interesting.

My major comments about this new version are:

-over all the manuscript, I noted inconsistencies between abbreviations for the PR locus (Pra; PR; PRA…); it is unclear if there are differences between these abbreviations;

-Polymorphism within population between the two PR alleles seems low to me for a region that has likely recombination suppression (L444:one single SNP); it doesn’t seem consistent with the dS value computed or I misinterpreted this part of the result.

Did the author use the denovo genome approach to check the polymorphism at PR loci as they did at HD loci?

-It is unclear if the hypothesis of a role in MAT genes for regulation of nuclear pairing is a new hypothesis or rely on previous observations (two references are cited L145; in the discussion, it appears as a new hypothesis) L678: indications could be given on how to further test this hypothesis.

-I had difficulties to understand the last part of the results about “shared nuclear haplotypes” and what the main conclusion of the author was. maybe a schema would help or clear conclusions.

-the text overall could be shorten (synteny analyses appear several times in the method, the part about STE3.2-1 loci could be merged in the result part)

Minor comments:

-L35: unclear what “genetic features” refers to

-L40: unclear what “allele status” refers to; bi-allelic or multiallelic

-L64: states it is an example

-L79: the model cited doesn’t states that outcrossing mating system is necessary in the model; the model may work under inbreeding or automixis.

-L84: cite Basidiomycota examples as they are more relevant to your study: Agaricus, Ustilago, Cryptococcus…

-L120 : what is known about the PR locus? It is explained L155, but could be explained here

-L125: the term evolutionary strata needs to be defined

-L143: unclear why the term “rudimentary” means

-L147-148: indicate on what evidence rely previous studies

-L195: use “degeneration” instead of “deterioration”

-L617 : no s at model if only one is cited

-L625: It is unclear to me “the locus is extended” ; do the author mean “the region with signal of recombination suppression around the PR locus”? how did the author define this region precisely ?

-L630-631: TE composition and coverage at the order level: not very clear to me what the order means

**Have all data underlying the figures and results presented in the manuscript been provided?**

Reviewer #1: Yes

Reviewer #2: Yes

Reviewer #3: Yes

PLOS authors have the option to publish the peer review history of their article (what does this mean?). If published, this will include your full peer review and any attached files.

Reviewer #1: **Yes: **Guus Bakkeren

Reviewer #2: No

Reviewer #3: No

---

## [Editor Report · Decision Letter 2]

28 Feb 2024

Dear Dr Schwessinger,

Thank you very much for submitting your Research Article entitled 'Genome biology and evolution of mating type loci in four cereal rust fungi' to PLOS Genetics.

The manuscript was fully evaluated at the editorial level and by independent peer reviewers. The reviewers appreciated the attention to an important topic but identified some concerns that we ask you address in a revised manuscript.

We therefore ask you to modify the manuscript according to the review recommendations. Your revisions should address the specific points made by each reviewer.

Yours sincerely,

Tatiana Giraud

Guest Editor

PLOS Genetics

Eva Stukenbrock

Section Editor

PLOS Genetics

The manuscript has been much improved. However, there remains multiple occurrence of awkward writing, including formulations already highlighted as incorrect in the previous rounds of reviews, as highlighted below.

-L4, L29, L213: mating-type loci

-L33/ as already said in the previous rounds of reviews, it’s not mating that is tetrapolar but the mating compatibility system

-L41 : we confirm that

-L38 : deleter extensive

-L44 : delete « of Puccinia » or « pathogenic Puccina species »

-L46 : delete « from sexual reproduction or somatic recombination » it does not make sense here

-L49 : sexes are, not sex is (sex is sexual reproduction)

-L50 : replace « allele divs » by « features », this sentence makes no sense

-L57-58, L210-211, 725-727: this is too far-fetched

-L61 : delete the ‘ signs ; delete loci or chromosome

-L71, 78, 80 : delete « or new TE insertions », it is unclear how this can suppress recombination here

-L72-76 : this is still very awkwardly explained and makes no sense in the present state.. replace by : « Instead, mathematical modelling and stochastic simulations suggest that non-recombining DNA fragments, caused for example by inversions, can be fixed solely due to the presence of deleterious mutations in genomes. Non-recombining fragments are beneficial if they carry fewer deleterious mutations than average and can increase in frequency. However, when becoming frequenct, their (few) recessive deleterious mutations will be exposed and selected against, preventing fixation. They can fix at permanent biallelic, heterozygous loci due to their sheltering of deleterious mutations (2). »

-L106, 108 and elsewhere: replace « mating behavior » by « mating compatibility system »

-L113 : bipolar mating compatibility, not behavior

-L115, 120, 160, 214, 215, 592 and elsewhere : tetrapolar mating compatibility, not behavior (you say L116 outcrossing behavior, which is here correct and shows the confusion around the term mating behavior in the crrent state)

-L117 : compatibility odds

-L157 : the mating compatibility system (compatibility would be between two cells)

-L179 : these studies did not « open » gaps, they brought insights and left gaps, or gaps remain after these studies

-L207 : at least three ? or is the fourth one multi-allelic ?

-Please stick to the accepted Latin abbreviation system for species names

-L362 : replace « aberrant patterns of gene or transposon density » by « particular patterns of gene or transposon density » (enriched TE patterns are not « aberrant » under recombination suppression)

-L431 : « whole-genome »

-L432 : genome singular as there is « respective » before

-L474 : « This species rank variation » : it is the level that is ranked not the variation itself

-L476, 497 : level instead of rank

-L543 : « the two MAT genes » : it sounds as you already know the two MAT genes while in the sentence you aim to investigate a potential mating-type function for a third one, so this sounds unclear, reformulate

-L593 : as already said, the causality is reverse : it’s not that outcrossing benefits from new alleles (which is wrong, group selection reasonning), but that, under outcrossing, new, rare allleles are selected for, this is completely different.

-L601 : unclea rformulation, add « each » after species ? or is the number all four species pooled ?

-L618 : this is also the case in Microbotryum, see Michael Hood’s studies

-L650 : « exact trends » seems contradictory, replace by patetrns

-L688 : why would they have other functions ? Delete this sentence, or rephrase saying there is no evidence of additional functions

---

## [Editor Report · Decision Letter 3]

4 Mar 2024

Dear Dr Schwessinger,

We are pleased to inform you that your manuscript entitled "Genome biology and evolution of mating type loci in four cereal rust fungi" has been editorially accepted for publication in PLOS Genetics. Congratulations!

Yours sincerely,

Tatiana Giraud

Guest Editor

PLOS Genetics

Eva Stukenbrock

Section Editor

PLOS Genetics

Comments from the reviewers (if applicable):

**Data Deposition**

http://datadryad.org/submit?journalID=pgenetics&manu=PGENETICS-D-23-00971R3

**Press Queries**

---

## [Editor Report · Acceptance letter]

12 Mar 2024

PGENETICS-D-23-00971R3 

Genome biology and evolution of mating type loci in four cereal rust fungi 

Dear Dr Schwessinger, 

We are pleased to inform you that your manuscript entitled "Genome biology and evolution of mating type loci in four cereal rust fungi" has been formally accepted for publication in PLOS Genetics! Your manuscript is now with our production department and you will be notified of the publication date in due course.

With kind regards,

Anita Estes

PLOS Genetics

On behalf of:
